



# Integrated ecological monitoring in Wales: the Glastir Monitoring and Evaluation Programme field survey

Claire M. Wood[1], Jamie Alison[2], Marc S. Botham[3], Annette Burden[2], François Edwards[3], R. Angus Garbutt[2], Paul B.L. George[4], Peter A. Henrys[1], Russell Hobson[5], Susan Jarvis[1], Patrick Keenan[1], Aidan M. Keith[1], Inma Lebron[2], Lindsay C. Maskell[1], Lisa R. Norton[1], David A. Robinson[2], Fiona M. Seaton[1], Peter Scarlett[3], Gavin M. Siriwardena[6], James Skates[7], Simon M. Smart[1], Bronwen Williams[2], Bridget A. Emmett[2]

[1] UK Centre for Ecology & Hydrology, Lancaster Environment Centre, Library Avenue, Bailrigg, Lancaster, LA1 4AP, UK
[2] UK Centre for Ecology & Hydrology, Environment Centre Wales, Deiniol Road, Bangor, Gwynedd, LL57 2UW, UK
[3] UK Centre for Ecology & Hydrology, Maclean Building, Benson Lane, Crowmarsh Gifford, Wallingford, Oxfordshire, OX1 08BB, UK
[4] School of Natural Sciences, Bangor University, Deinol Road, Bangor, Gwynedd, LL57 2UW, UK
[5] Butterfly Conservation, Manor Yard, East Lulworth, Wareham, Dorset, BH20 5QP, UK
[6] British Trust for Ornithology, BTO, The Nunnery, Thetford, Norfolk IP24 2PU, UK
[7] Welsh Government, Sarn Mynach, Llandudno Junction, Conwy, UK

*Correspondence to*: Claire M. Wood (clamw@ceh.ac.uk)

**Abstract.** The Glastir Monitoring and Evaluation Programme (GMEP) ran from 2013 until 2016, and was probably the most comprehensive programme of ecological study ever undertaken at a national scale in Wales. The programme aimed to (1) set up an evaluation of the environmental effects of the Glastir agri-environment scheme and (2) quantify environmental status and trends across the wider countryside of Wales. The focus was on outcomes for climate change mitigation, biodiversity, soil and water quality, woodland expansion and cultural landscapes. As such, GMEP included a large field survey component, collecting data on a range of elements including vegetation, land cover and use, soils, freshwaters, birds and insect pollinators from up to 300 1 km squares throughout Wales. The field survey capitalised upon the UKCEH Countryside Survey of Great Britain, which has provided an extensive set of repeated, standardised ecological measurements since 1978. The design of both GMEP and the UKCEH Countryside Survey involved stratified-random sampling of squares from a 1 km grid, ensuring proportional representation from land classes with distinct climate, geology and physical geography. Data were collected from different land cover types and landscape features by trained professional surveyors, following standardised and published protocols. Thus, GMEP was designed so that surveys could be repeated at regular intervals to monitor the Welsh environment, including the impacts of agri-environment interventions. One such repeat survey is scheduled for 2021 under the Environment and Rural Affairs Monitoring and Modelling Programme (ERAMMP).

Data from GMEP have been used to address many applied policy questions, but there is major potential for further analyses. The precise locations of data collection are not publicly available, largely for reasons of landowner confidentiality. However,





the wide variety of available datasets can be (1) analysed at coarse spatial resolutions and (2) linked to each other based on

square-level and plot-level identifiers, allowing exploration of relationships, trade-offs and synergies.

This paper describes the key sets of raw data arising from the field survey at co-located sites, 2013 to 2016. Data from each

of these survey elements are available with the following Digital Object Identifiers. Landscape features,

https://doi.org/10.5285/82c63533-529e-47b9-8e78-51b27028cc7f, https://doi.org/10.5285/9f8d9cc6-b552-4c8b-af09-

e92743cdd3de, https://doi.org/10.5285/f481c6bf-5774-4df8-8776-c4d7bf059d40 ; Vegetation plots,

https://doi.org/10.5285/71d3619c-4439-4c9e-84dc-3ca873d7f5cc ; Topsoil physico-chemical properties,

https://doi.org/10.5285/0fa51dc6-1537-4ad6-9d06-e476c137ed09 ;

Topsoil meso-fauna, https://doi.org/10.5285/1c5cf317-2f03-4fef-b060-9eccbb4d9c21; Topsoil particle size distribution

https://doi.org/10.5285/d6c3cc3c-a7b7-48b2-9e61-d07454639656 ; Headwater stream quality metrics,

https://doi.org/10.5285/e305fa80-3d38-4576-beef-f6546fad5d45 ; Pond quality metrics, https://doi.org/10.5285/687b38d3-

2278-41a0-9317-2c7595d6b882 ; Insect pollinator and flower data, https://doi.org/10.5285/3c8f4e46-bf6c-4ea1-9340-

571fede26ee8; Bird counts, https://doi.org/10.5285/31da0a94-62be-47b3-b76e-4bdef3037360.

## 1 Introduction

The Welsh Government initiated the Glastir Monitoring and Evaluation Programme (GMEP) in 2013 to evaluate the

environmental effects of the Glastir agri-environment scheme, but also to monitor the wider countryside of Wales (Emmett et

al., 2015). In Wales, funding from agri-environment schemes (AES) has been available since the early 1990s including

Environmentally Sensitive Areas (ESAs), the Habitat Scheme, Woodland Grant scheme, Farm and Conservation grant scheme,

Tir Cymen, Tir Cynnal, Tir Gofal and most recently, Glastir. Currently, the Glastir scheme is the main method that the Welsh

Government pays for environmental goods and services (Emmett and GMEP team, 2014). The primary aim of GMEP

monitoring was to collect evidence for the effectiveness of bundles of management interventions in delivering outcomes of

interest related to climate change mitigation, biodiversity, soil and water quality, woodland expansion and cultural landscapes.

Two additional objectives for reporting were added by the Auditor General for Wales in 2014: (1) to increase the level of

investment in measures for climate change adaptation, with the aim of building greater resilience to ongoing climate change

into both farm and forest businesses and the wider Welsh economy; (2) to use agri-environmental investment in a way that

contributes towards farm and forest business profitability and the wider sustainability of the rural economy (Emmett and

GMEP team, 2017).

The monitoring also collected evidence to quantify the status and trends in the environment in general and contributes to State

of the Natural Resources Reporting (SoNaRR) (Natural Resources Wales, 2020). The data collected may be analysed in order

to identify how drivers of change, such as land use, climate and pollution affect the Welsh environment, beyond Glastir



interventions (Emmett and GMEP team, 2014). This paper describes the key sets of raw data arising from the field survey element of GMEP, undertaken between 2013 and 2016.

## 1.1 Introduction to the GMEP Survey Design

While GMEP encompassed a range of different components, including modelling and socio-economic surveys, a field survey
formed the largest element of the monitoring programme. The field survey was designed in such a way as to capture multiple measures and metrics, and to integrate across these metrics. In order to do this, a full ecosystem-based approach was chosen such that data were captured across multiple scales, where possible during a single field visit. A 4-year cycle rolling survey was adopted in order to maximise the number of sites visited at the national scale, while also monitoring year-on-year. This would allow cost-effective detection of both spatial variation and temporal trends (Emmett and GMEP team, 2014). The first
survey cycle dates from 2013 to 2016, with the potential for, and intention of, regular repeat surveys.

Across GMEP monitoring, integration of survey data was a priority and therefore a common spatial unit of 1 km square was adopted. A total of 300 1 km squares (Fig. 1), were sampled over the four year cycle. 1 km squares were a conveniently sized unit for landscape monitoring which has been adopted by previous successful monitoring programmes. First tested for this type of monitoring in a small scale survey in Cumbria (1975) (Bunce and Smith, 1978) and Shetland (1974) (Wood and Bunce,
2016), the 1 km monitoring unit was later adopted for the Countryside Survey of Great Britain from 1978 (Bunce, 1979) to present day and is used by other current monitoring schemes, such as the Breeding Bird Survey (Harris et al., 2018) and the Wider Countryside Butterfly Survey (Brereton et al., 2011). The 300 GMEP field survey squares were split evenly into two key components: the 'Wider Wales Component' used for baseline estimation, national trends and national reporting of Glastir, and the 'Targeted Component', which focussed on priority areas and aims of the Glastir scheme (Emmett and GMEP team,
85   2014).

The 'Wider Wales Component' of GMEP comprised 150 1 km squares which were selected following the same procedure as used for the UKCEH Countryside Survey of Great Britain (Carey et al., 2008; Norton et al., 2012), aiming to provide statistically robust estimates of indicators from 1978-2016 at national and sub-national levels. Thus, 'Wider Wales' squares were a stratified-random sample of Wales, with proportional representation of strata defined according to the ITE Land
Classification of Great Britain (henceforth "land classes") (Bunce et al., 2007; Emmett and GMEP team, 2014). Land classes are derived from a statistical analysis of topographic, physiographic, geological and climatic attributes. Environmental heterogeneity is minimized within each land class, and is maximised between land classes. The number of 1 km squares randomly sampled from each land class was proportional to the area of that land class in Wales. This helped to optimise allocation of survey effort (Emmett et al., 2015).

The other half of the sampled squares were targeted specifically at Glastir priority areas (Welsh Government, 2020). The squares were selected by calculating weights for each 1 km square across Wales that reflected the amount and diversity of Glastir uptake within the square (Emmett and GMEP team, 2014). Squares were then randomly selected with probability proportional to these assigned weights, such that a square with twice the weight as another was twice as likely to be selected.



The weighting, and therefore selection of 'Targeted Component' squares, was repeated each year as new information became
available on Glastir uptake. Across both GMEP field survey components, any square that contained more than 75 % of urban
land or that was more than 90 % sea (defined by the UK Land Cover Map 2007 (Morton et al., 2011) and mean high tide data
(Ordnance Survey, 2020)) was excluded and replaced according to the above procedures (Emmett and GMEP team, 2014).

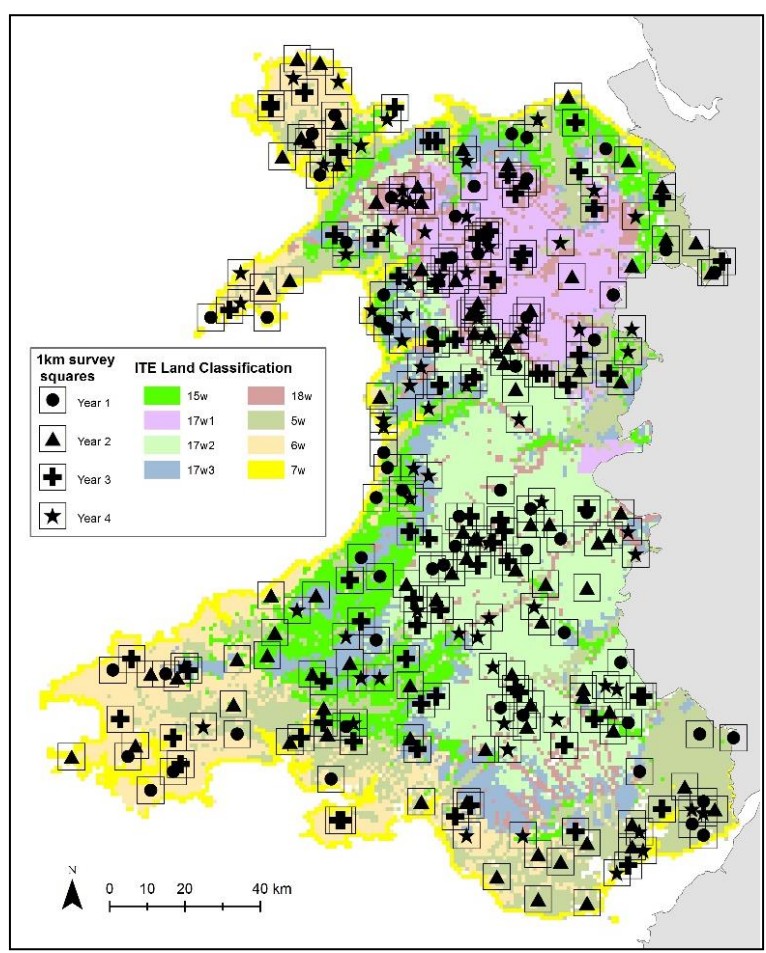


**Figure 1. Map to show distribution of 1 km survey squares across different land classes in Wales, (survey squares not shown to
scale, to preserve data confidentiality).**

## 2 Data collected: field and laboratory collection methods

A wide range of data were collected during the field survey, encompassing land use and cover, vegetation, soils, freshwater,
birds and insect pollinators. Fig. 2 illustrates the type and distribution of data collected in a typical 1 km survey square and a

short summary of each of the elements is provided in this section. In addition to these key, published datasets, ancillary information was collected at survey sites regarding landscapes (photographs), footpath and historic feature assessments.


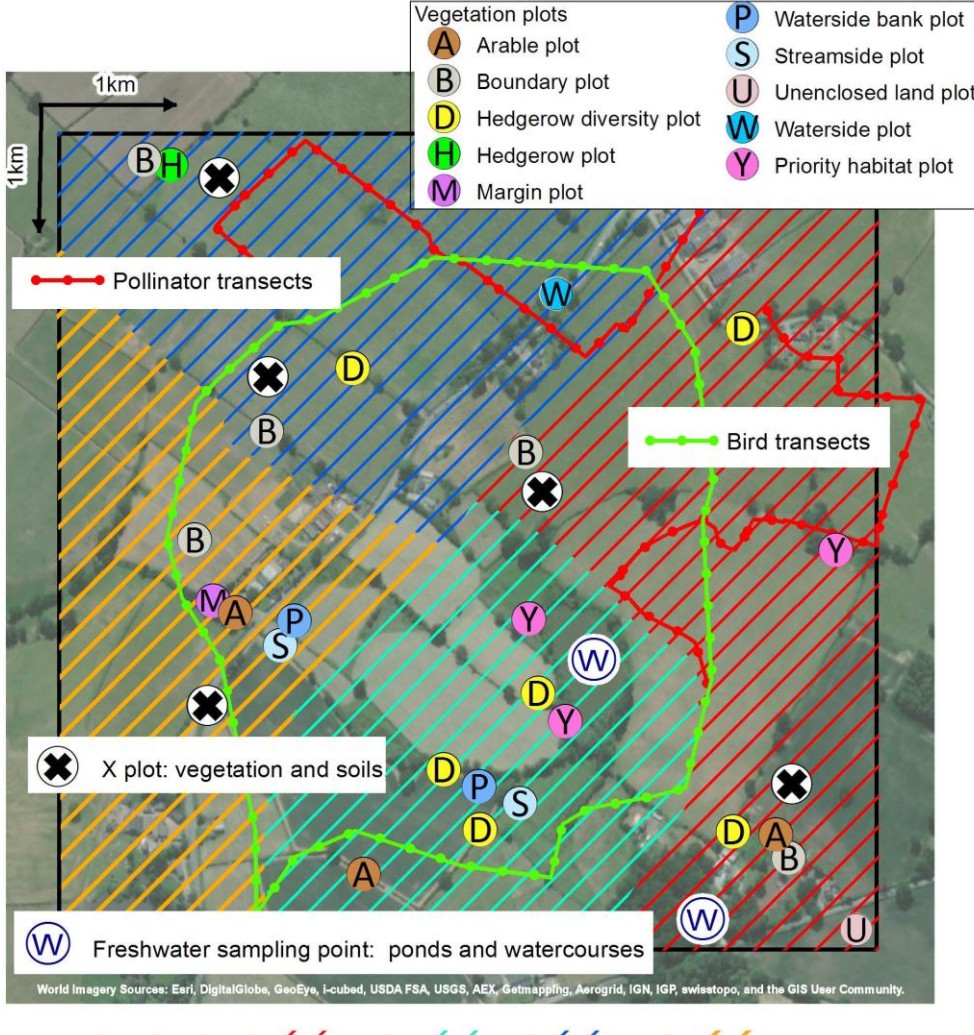

**Figure 2. Type and distribution of published data collected in a typical 1 km survey square (excluding the land cover/land use mapping element, see Fig. 3.) (Esri, 2021)**



## 2.1 Land Cover/Land Use

The most geographically comprehensive element of the survey is the mapping of land cover and ecologically relevant landscape features (Fig. 3). The methods adopted were those of the UKCEH Countryside Survey, described in detail in Wood et al. (2018a). Across accessible areas of each 1 km square, areal, linear and point features were mapped digitally using Windows 7 (Microsoft) based electronic data capture equipment, and electronic mapping software ("CS Surveyor") co-developed by the UK Centre for Ecology & Hydrology and software company, Esri UK (Maskell et al., 2008). With the aid of

base maps, each feature was assigned a range of pre-determined coded attributes (Maskell et al., 2008; Wood et al., 2018a). For area features, attributes for each mapped polygon included Biodiversity Action Plan (BAP) broad/priority habitats (Jackson, 2000; Maddock, 2008), land use and land management (for example, crop, grazing animals, recreation, timber, burning), dominant vegetation species, and a variety of other descriptors according to the land use type (for example, road verge widths, tree diameter at breast height, woodland structure, woodland features and sward descriptions) (Wood et al.,

2018b).

Linear features are landscape elements less than 5 m wide that form lines in the landscape (Wood et al., 2018b). Recording included the length and condition of a range of linear features predominantly, but not exclusively, describing boundaries. These include managed woody linear features (i.e. hedges), unmanaged woody linear features (i.e. lines of trees), walls, fences, streams and a range of other linear features. Recorded linear features have a minimum length of 20 m and may include gaps

of up to 20 m. All linear features were recorded unless they form part of a curtilage or they are within the woodland canopy. Woody linear features, including hedges, remnant hedges and lines of trees were classified using a key (Maskell et al., 2016a) following consultation with the Hedgerow Steering Group of the UK BAP (Wood et al., 2018b).

Point features are individual landscape elements that occupy less than an area of 20 m × 20 m. Point features may be trees or groups of trees, ponds and other freshwater features, physiological features such as cliffs, buildings and other structures with

various use codes (for example, "residential" or "agricultural") (Wood et al., 2018b). For the detailed methodology, see the GMEP Field Mapping Handbooks (Maskell et al., 2016a; Maskell et al., 2016b). Quality assurance was achieved by ensuring surveyors were trained appropriately before each field season, visits to surveyors in the field by supervisors, and the repeat survey of a number of squares to identify any issues arising.


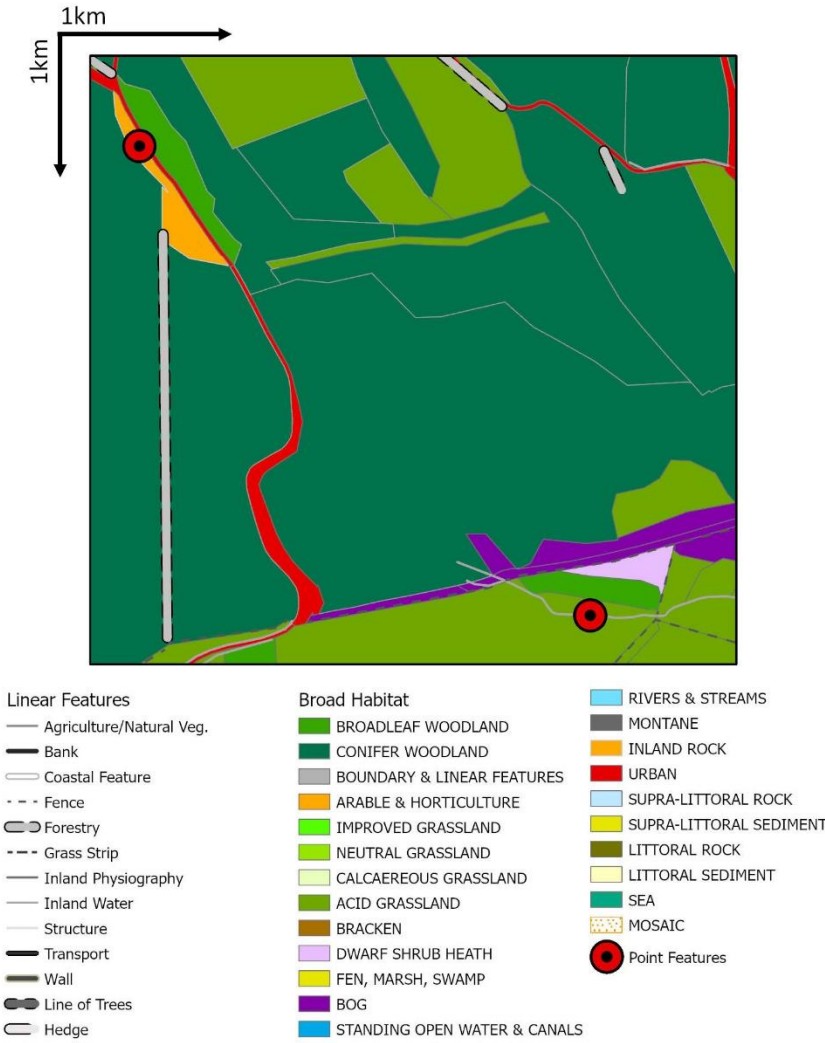

**Figure 3. An example of a survey square, showing mapped point, line and area features (key includes the full range of possible broad habitats and linear features, not all shown on the map).**

## 2.2 Vegetation Plots

The vegetation element of the field survey involved recording plant species presence and cover in different sizes and types of vegetation plot (Table 1) comprising different numbers of "nests" (i.e. sub-sections of the plot). The design of the plots originated in the UKCEH Countryside Survey, and the history and logic behind their positioning is described fully in Wood et al. (2017). A comprehensive description of these plots may be found in the field survey handbook (Smart et al., 2016) with a



summary presented in Table 1. In each vegetation plot, a complete list of all vascular plants and a selected range of readily identifiable bryophytes and macro-lichens was made, with the exception of 'D' (Diversity) plots, in which only woody species in hedgerows were recorded (Wood et al., 2017). Cover estimates were made to the nearest 5 % for all species reaching at least an estimated 5 % cover. Presence was recorded if cover was less than 5 %. Canopy cover of overhanging trees and shrubs was also noted, even if individuals were not rooted within the plot. Additionally, general information about the plot was recorded to provide supporting information for relocation and analytical purposes as well as describing potential habitats such as glades

and dead wood. To ensure quality, the field training courses held before the surveys covered identification of difficult species, regular visits were made to survey teams by managers, and difficult specimens could be collected and sent to experts for identification (Wood et al., 2017). However, predetermined combinations of species may have been recorded as aggregates reflecting known difficulties in their separation in the field. A number of plots were repeated by quality assessors to ensure consistency of quality within the survey (Wood et al., 2017).


**Table 1. Plot types included in GMEP (Smart et al., 2016)**

| Code | Name | Where | Size | No. per 1km square | Additional information *(sampling followed standard protocols established by the UKCEH Countryside Survey www.countrysidesurvey.co.uk unless stated below)* |
|---|---|---|---|---|---|
| **X** | ***Random/Main/ X-plot*** | *Dispersed random points (not on linear features)* | 4 m$^2$ or 200 m$^2$ | Up to 5 | In years 2013-2014, all X plots were 200 m$^2$. In 2015-2016, due to resource limitations, plots were reduced to 4m$^2$ (with the exception of a small sub-set of squares). |
| **Y** | ***Small: Targeted and Enclosed/Habitat*** | *Primarily allocated to 'enclosed habitats' in or out of Glastir option and then additionally placed to record priority habitats (PH) not sampled by other plots.* | 4 m$^2$ | Up to 5 but more if >5 PH | * |
| **U** | ***Unenclosed*** | *Unenclosed broad habitats in or out of Glastir options.* | 4 m$^2$ | Up to 10 | * |
| **B** | ***Boundary*** | *Adjacent to field boundaries in or out of Glastir option.* | 10 x 1 m | 5 | * |
| **A** | ***Arable*** | *Arable field edges centred on each B plot. In or out of Glastir option but only one per arable field. Paired with X plots if out of option.* | 100 x 1 m | Up to 5 | * |
| **M** | ***Margin*** | *Field margins in or out of Glastir option.* | 2 x 2 m | Up to 15 | * |
| **H** | ***Hedgerow*** | *Alongside hedgerows (i.e. WLF with unnatural shape) and usually coincident with two of the D plots.* | 10 x 1 m | 2 | |





| D | *Hedgerow diversity* | *WLF with natural or unnatural shape. Allocated proportionally to WLF in Glastir option.* | 30 x 1 m | Up to 10 | * |
| S/W | *Streamside* | *Four placed alongside watercourses and allocated in proportion to Glastir option uptake. One W plot centred on the RHS stretch.* | 10 x 1 m | Up to 5 | * |
| P | *Perpendicular streamside* | *Sampling the upslope habitats adjacent to and centred on the S/W plots.* | 10 x 1 m | Up to 5 | A new type of plot for the GMEP survey. Nests within these plots are of variable length, summing to 10 m (Emmett et al., 2015). |
| R/V | *Roadside verge plots* | *Sampling the 1m strip adjacent to roads and tracks.* | 10 x 1 m | Up to 5 | Only recorded in 2016, in a sub-set of squares. |

*Location of these plots incorporated additional targeting to take into account the proportional amount of Glastir options in a 1 km survey square.*


### 2.3 Soils

Within each of the 300 x 1 km sample squares, the key soil measurements described below were taken from a set of three volumetric topsoil samples (0-15 cm) sampled from each of five pre-determined randomly dispersed locations, using standard
sized plastic tubes and a metal coring implement. The sampling locations were coincident with the five 'Main'/'X'/large vegetation plots as described above (Table 1). Sampling started on the southern corner of the inner 2 x 2 m nest of the X plot in 2013, then west, north and east in the consecutive years. Soil samples included: one sample analysed for physico-chemical soil metrics, taken using a black plastic core (15 cm long x 5 cm diameter); a spare white core (15 cm long x 5 cm diameter), and a sample for soil fauna (2013 and 2014), taken using a shorter core (8cm long). In 2013 and 2014, five bulked 0-15 cm
gouge auger samples for DNA metabarcoding were also taken and were frozen upon receipt at the laboratory until analysis (George et al., 2019a). After collection, the soil cores were refrigerated and stored until posted, usually within two days, to laboratories at the UK Centre for Ecology & Hydrology in Bangor and Lancaster for analysis and/or archive storage in air-dried or frozen form.

### 2.3.1 Physico-chemical properties

The sample taken for analysing physico-chemical properties included measurements of the following properties: Loss-on-ignition (LOI) and derived carbon concentration, total soil organic carbon (SOC) and nitrogen, total soil phosphorous, Olsen-phosphorous, soil pH (in deionized water and calcium chloride), soil solution electrical conductivity, soil bulk density of fine
earth, fine earth volumetric water content (where sampled), soil water repellency and water drop penetration time. The methods for these are summarized in Table 2.





**Table 2. Summary of soil physico-chemical properties and measurement methods**

*i. Loss-on-ignition*

Loss-on-ignition (LOI) is a simple and inexpensive method for determining soil organic matter and estimating soil organic carbon concentration using an appropriate conversion equation that can be determined using total carbon analysis. The method was the same standard method as that used in the UKCEH Countryside Survey (Emmett et al., 2010). LOI was measured on a 10 g air dried sub-sample taken after sieving to 2 mm. For samples where less than 10 g was available, as close as possible weight of soil was used and records of the exact quantity were recorded. The sub-sample was dried at 105 °C for 16 hours to remove moisture, weighed, then combusted at 375 °C for 16 hours. The cooled sample was then weighed, and the LOI (%) calculated (Emmett et al., 2010). In order to have data that were compatible with legacy data from other surveys, like UKCEH Countryside Survey that emphasised the measurement of LOI, carbon (C) concentration was derived from the LOI measurement. Resulting C concentration measures, unlike those from some other methods, were thus unaffected by soil inorganic carbon.   The formula for deriving C concentration is:

$$\text{C concentration (g C kg}^{-1}) = \text{LOI (\%)} * 0.55 * 10$$

LOI quality control checks were carried out using internal soil standards prepared in an identical manner to the sampled soils. Two different internal standards were included in each sample batch. Those internal standards were compared with an historically generated mean value for internal standards. If the measured LOI for the two internal standards in a batch varied by more than 2 standard deviations, in either direction, from the historic mean value, then the batch was repeated.

*ii.        Total soil organic carbon (SOC) and nitrogen*

This analysis was carried out using the UKAS accredited method SOP3102, at UKCEH Lancaster.  Soil samples were air dried (at 40 °C), ball milled and oven dried at 105°C (± 5°C) for a minimum of 3 hours. Samples were then analysed using an Elementar Vario-EL elemental analyser (Elementaranalysensysteme GmbH, Hanau, Germany). The Vario EL is a fully automated analytical instrument working on the principle of oxidative combustion followed by thermal conductivity detection. Following combustion in the presence of excess oxygen, the oxides of nitrogen (N) and carbon (C) flow through a reduction column which removes excess oxygen. C is trapped on a column whilst N is carried to a detector. C is then released from the trap and detected separately. Sample weights are usually 15 mg for peat and 15-60 mg for mineral soil samples (Emmett et al., 2010). Quality control was achieved by use of two in-house reference materials analysed with each batch of samples.

*iii.       Total soil phosphorous (P)*

Air-dried and ground (to 2 mm) soils were digested with hydrogen peroxide (100 Volumes) and sulphuric acid in a 5:6 ratio. Selenium powder and lithium sulphate were added to raise the boiling point of the acid.

Samples were then placed at 250 °C for 15 minutes and then to 400 °C where the temperature was maintained for 2 hours to complete the digestion. After digestion, the samples were diluted with ultrapure water and allowed to settle overnight. The supernatant was then further diluted and P was measured colourimetrically using a SEAL AQ2 discrete analyser. Phosphorus was determined using ammonium molybdenum blue chemistry with the addition of ascorbic acid to control the colour production. Two quality control reference samples, a duplicate sample and two matrix matched blanks were run every 25 samples to ensure data quality. The final concentration (mg/Kg) was determined using a calibration curve of the standard and took into account the blank concentration.



### iv. Olsen-phosphorous

Olsen-phosphorous was measured in samples from arable and improved grassland habitats only, where the measurement is most reliable (Emmett et al., 2010). Two grams of air dried soil samples were extracted in 40 ml Olsen's reagent (0.5 M NaHCO3 at pH 8.5) for 30 minutes in a mechanical end-over-end shaker. The sample was then filtered through a Whatman 44 filter paper to separate the soil and the filtrate; the filtrate is kept for analysis. The analysis was performed on a Seal Analytical AA3 segmented flow. The samples were mixed in the flow channel with an acidic ammonium molybdate and potassium antimony tartrate to form a complex with phosphate. This complex was reduced with ascorbic acid to develop a molybdenum blue colour. The reaction was temperature controlled to 40°C using a water bath to ensure uniform colour development. The developed colour was measured at 880 nm.

Two quality control reference samples, a duplicate sample and two blanks were run every 25 samples to ensure data quality. The final concentration is expressed in milligrams per kilogram (mg/kg) and is for moisture content, the concentration of the blank and using a calibration curve of the standard.

### v. Soil pH in deionized water and calcium chloride (CaCl$_2$)

Soil pH was carried out on a suspension of fresh field-moist soil in deionised water and 0.01 M CaCl$_2$. The ratio of soil to water or CaCl$_2$ was 1:2.5 by weight. The method used was based upon that employed by the Soil Survey of England and Wales (Avery and Bascomb, 1974). Two different internal standards were included in each sample batch for quality control. Batches in which the measured pH for the internal standards varied by more than 2 standard deviations in either direction from the mean value generated historically for the internal standards were repeated.

### vi. Soil solution electrical conductivity

10 g of field moist soil was weighed into a beaker with 25 mL of deionised water added, then stirred with a rod to produce a homogeneous suspension. After half an hour, the contents of the beaker were stirred again with the rod, and the electrical conductivity (EC) was measured using an electrode and a conductivity meter (Jenway 4510). For quality control, two different internal standards were included in each sample batch. Those internal standards were compared with an historically generated mean value for internal standards. If the measured EC for the two internal standards in a batch varied by more than 2 standard deviations, in either direction, from the historic mean value, then the batch was repeated.

### vii. Soil bulk density of fine earth and volumetric water content of fine earth

The bulk density (BD) of soil depends greatly on the mineral make up of soil, soil organic matter and the degree of compaction. The density of quartz is around 2.65 g/cm³ but the (dry) bulk density of a mineral soil is normally about half that density, between 1.0 and 1.6 g/cm³. Soils high in organic content and some friable clay may have a bulk density well below 1 g/cm³. Bulk density is a measure of the amount of soil per unit volume. It is therefore an excellent measure of available pore space in a soil, and gives information on the physical status of the soil. BD values are also essential when estimating soil C stocks, as they allow a conversion from %C to C per unit volume. Bulk density was determined from a core which is 15 cm long with a diameter of 5 cm. Dry bulk density is calculated using the following equation:

$$\text{Dry Bulk Density (g cm}^{-3}) = \frac{(\text{Dry weight core (105 °C) (g) - stone weight (g)})}{(\text{Core volume (cm}^{-3}) \text{ - stone volume (cm}^{-3})}$$

### viii. Fine earth volumetric water content when sampled



Once the bulk density was calculated, the volumetric water content of the fine earth fraction could be determined by multiplying the bulk density and the gravimetric water content of the fine earth. Quality control was achieved by using fixed volume pre-cut sleeves for soil sampling and extensive training for soil surveyors.

### ix.  *Soil water repellency, water drop penetration time*

Soil water repellency (surface) measurement was carried out by measuring the time for a fixed volume droplet of deionised water (100 uL) to be fully absorbed into the soil surface (water drop penetration time (WDPT)). 6 drops of water were applied to an air dried undisturbed soil surface. The entire process was filmed using a digital video camera so that the timing could be determined accurately. The samples were maintained in a laboratory at a relatively constant temperature ~20 ºC. Some soils, especially arable, were not consolidated so measurements were taken on surface unconsolidated soil or aggregates using 20 g soil added to a tin lid, and procedure followed as described above. For quality control, a micropipette was used to deliver the drops, 6 drops were used and then the median value obtained. All drop penetration was captured using video so that times of penetration can be reviewed if required.


### 2.3.2 Soil meso-fauna

Soil meso-fauna were extracted using the standard Tullgren-funnel method (Southwood, 1994), as used in the UKCEH Countryside Survey (Emmett et al., 2008). Each Tullgren funnel unit (Burkard Scientific Ltd, Uxbridge, UK) consists of an aluminium funnel with a rubber seal to attach a collection bottle during the extraction period and an aluminium sieve on top.

Each unit is positioned (through a hole) beneath a light bulb holder. Soil cores are placed in the sieve with the bulb turned on to extract soil fauna over an extended period; typically 5 days. A 40 W bulb provides heat to drive the soil fauna from the soil core into collection bottles filled with preservative; usually 70 % ethanol. Following this extraction procedure, the samples were sorted, identified and enumerated according to broad groups (as shown in Table 3) by trained staff and students (Emmett et al., 2008). These enumerated broad groups were entered onto data template spreadsheets and once complete sent to UKCEH

Bangor for integration into the database.

**Table 3. Soil meso-fauna**

| | Enumerated broad groups | | Details |
|---|---|---|---|
| 1 | *Acari* | *Oribatid - Phthiracaridae* | Commonly known as 'Box' mites; these decomposers tend to be abundant in woodlands. |
| 2 | | *Oribatid* - Others | Other oribatid mites, mostly decomposers or microbial feeders. |
| 3 | | *Mesostigmatid* | Predatory mites which feed on other soil mesofauna. |
| 4 | | Other | Typically small mites; largely containing prostigmatids and juvenile mesostigmatids. |
| | | | |
| 5 | *Collembola* | *Poduromorpha* | Poduroid Collembolans; short legs and plump body shape. |




| 6 | *Entomobryomorpha* | Entomobryoid Collembolans; generally with long, slender body. |
| 7 | *Symphypleona/Neelipleona* | Symphypleonid Collembolans; Small, round, globular body shape. |
| 8 | Total oribatids | 1+2 |
| 9 | Total mites | 1+2+3+4 |
| 10 | Total Collembolans | 5+6+7 |
| 11 | Total mesofauna | 1+2+3+4+5+6+7 |

For the purpose of quality control, another member of staff checked 1 in 20 samples for the first 200 samples. Fauna were then identified and enumerated by both members of staff to ensure that the identification and counting procedures employed by both individuals produced comparable results. This process was repeated at a reduced rate as the identifications proceeded (Emmett et al., 2008).

**2.3.3 Particle size distribution (PSD)**

The particle size distribution (PSD) of a soil, typically presented as the proportions of clay (<2 µm), silt (2-63 µm) and sand (63-2000 µm), is a fundamental property of the soil. It controls nearly all edaphic processes and exerts strong control on hydrology, transport of pollutants, availability of nutrients, stabilization of soil organic matter, mechanisms of erosion, gas

exchange, soil biota and above ground productivity. The method of laser diffraction (LD) emerged in the 1980s as a potentially powerful tool for analysing granular materials and in the 1990s, the soil science community began to apply LD to soils (e.g. (Lebron et al., 1993)). The method has the advantage of being quick (about 5 minutes per sample), requires small amounts of soil (<2.0 g), is reproducible and provides a wide range of size classes (rather than the conventional 3 to 9).

For GMEP, particle size distribution was analysed in samples with a loss-on-ignition lower than 50 % using a Beckman Coulter

LS13 320 laser diffraction particle size analyser (Beckman Coulter Inc.) and the hydrometer method (Gee and Or, 2002) (Emmett et al., 2015).

Before running the LD instrument, it was rinsed and calibrated and a sample of sand was run to flush any microbiological growth from the system. Standard soil samples were included with each batch of samples and duplicated samples were included (one in ten) to check for reproducibility. To evaluate the accuracy of the instrument, different sized standards were used:

nominal 500 µm glass beads (Beckman Coulter Inc.), nominal 15 µm Garnnet (Beckman Coulter Inc.). Sandy soil from Gleadthorpe (Cuckney, UK), clay soil from Brimstone (Denchworth, UK) and a silty soil from Rosemaud (Bromyard, UK) were also used. All three soils are well-characterised farm soils from ADAS Ltd. In addition, two well-characterised internal soil standards from the UKCEH laboratory were used (loam and silty clay loam).

The sand fraction was collected with a 63 µm sieve at the end of the drainage outlet. As a way to corroborate the laser

measurements, the weight of the sand collected in the sieve at the end of the measurement was compared with the data provided



by the instrument. In general, there was a good agreement for both values for the sand fraction. However, high content of organic matter interference with the laser measurements was observed. After removal of organic matter, when the soil is very organic (loss on ignition (LOI) values of 40-50%), there are still some recalcitrant organic materials that persist in the soil and produce overestimation of the sand fraction measure with LD.


## 2.4 Freshwaters

A range of different types of data were collected from the freshwater habitats of headwater streams and ponds. Data were collected across the 300 1 km survey sites, where the features occurred.


### 2.4.1 Headwater streams

Data were collected from selected sections of headwater streams where present in up to 300 x 1km squares according to standardised field methods (Kelly et al., 1998; Murray-Bligh, 1999; O'Hare et al., 2013). Sampling points were generally

chosen to enable a full River Habitat Survey (River Habitat Survey, 2021) to be taken in the square, while also being as close as possible to an access point into the square. Data relating to this freshwater element of the survey are summarised in Table 4.

**Table 4. Summary of freshwater properties and measurement methods**


| i. | Diatoms |
| --- | --- |
| Samples were digested using hydrogen peroxide to remove organic matter and mounted on slides using the mountant Naphrax. At least 300 valves on each slide were identified to the highest resolution possible using a Nikon BX40 microscope with 100x oil immersion objective with phase contrast. The primary floras and identification guides used were Krammer and Lange-Bertalot (1986); Krammer and Lange-Bertalot (2000); Krammer and Lange-Bertalot (1997, 2004), Hartley (1996) and Hofmann et al. (2011). All nomenclature was adjusted to that used by Whitton et al. (1998) which follows conventions in Round et al. (2007) and Fourtanier and Kociolek (1999). Members of the *Achnanthidium minutissimum* complex showed considerable morphological variability and were classified using the conventions in Potapova and Hamilton (2007). | |
| i. | Invertebrates |
| Initially, preservative and silt were washed from the samples in a fume cupboard by rinsing repeatedly with tap water. The samples were split into smaller portions to enable thorough washing. A series of sieves, reducing in coarseness, could also be used. The samples were finally sieved through a 500 µm sieve (mandatory). Samples with a large quantity of filamentous algae or plant material were floated in a bucket of water so the trapped invertebrates fell to the bottom. | |
| Small portions of the samples were placed into a water filled sorting tray, marked with a grid to act as an aid, and systematically scanned for invertebrates. Large pieces of detritus were removed and the invertebrates were then removed using forceps or tweezers. Examples of | |





all taxa were placed into vials for quality assurance. Specimens may have been identified straight away or separated into different taxon groups for more detailed investigation. Large clumps of vegetation were broken down to ensure that no specimens were overlooked. After the first sort, the sample was then disturbed and/or rotated to expose previously hidden taxa. The process was repeated until all necessary taxa had been removed from the sample.

Macroinvertebrates removed from the sample were identified to species level where possible, including Caddis and Diptera pupae but with the exception of *Oligochaeta*, *Chironomidae*, *Simuliidae* and *Hydracarina*. The majority of identification work was conducted using dissection microscopes with high power microscopes used for examination of small specimens or specific parts of specimens. Terrestrial and aerial stages of aquatic species, terrestrial species and specimens which were dead when collected were not counted. Invertebrates which had become fragmented were only counted as a record if the thorax and abdomen were present. If only the posterior, abdomen or head were present the species was not recorded.

Though the abundances of taxa were not necessary for the Biological Monitoring Working Party (BMWP) water quality scoring system, they were recorded for use in other indices and environmental diagnostics. Only free-living individuals were counted. Colonies were counted as one individual. Very abundant taxa were recorded by distributing the sample evenly, counting the specimens in a portion of the tray using the grid lines and calculating the total by proportions. Quality control was carried out by the re-analysis of one randomly selected sample per twenty by a different analyst.

### i.        Water chemistry

Bottled water samples were sent to the laboratories at the UK Centre for Ecology and Hydrology, Lancaster and were analysed according to accredited methods as below:

*Phosphate (PO4-P):*

$PO_4$-P concentrations were measured colourimetrically using a Seal Analytical AQ2 discrete analyser. $PO_4$-P was determined by reaction with acidic molybdate in the presence of antimony to form an antimony-phospho-molybdate complex. Ascorbic acid reduced this to the intensely blue phosphomolybdenum complex, measured spectrophotometrically at 880nm. Calibration was produced by automatic dilution of a single stock solution of 0.2 mg/l $PO_4$-P; concentrations were obtained using the calibration curve within the range 0 – 0.2 mg/l. Control standards of 0.1 mg/l $PO_4$-P were analysed every 10 samples.

*Total dissolved nitrogen (TDN):*

A Skalar Formacs CA16 analyser with an attached ND25 was used to measure total dissolved nitrogen in water samples. TDN was measured by combustion at 900 °C with a cobalt chromium catalyst which converts all nitrogen to nitric oxide. The nitric oxide was measured by a chemi-luminescent reaction with ozone. The calibration range of the instrument was 0-4 mg/L for nitrogen. Samples with values over these were diluted within range using 18.2 MΩ carbon free water.

*Alkalinity:*

Alkalinity was determined using a standard operating procedure for alkalinity in waters. Alkalinity was determined using the Mettler Toledo DL53 titrator which performs analyses automatically using predefined methods. A complete titration method comprised sample dilution, dispensing of acid, stirring and waiting times, the actual titration, the calculation of results and a report.

Reagents and material used include: standard buffer solutions from Fisher Scientific of pH 4 and pH 7, electrode filling solution made with potassium chloride solution (4 M) saturated with silver chloride from Fisher Scientific, 1.0 M Hydrochloric acid, 0.02 M Hydrochloric acid, 1000 mg/l stock solution as calcium carbonate, 20 mg/l (400 ue/l) QC standard as calcium carbonate, deionised water.



Alkalinity (usually) reflects the activity of calcium carbonate, so results were reported as milligrams per litre of calcium carbonate (mg/l CaCO3).

### 2.4.2 Ponds

A bottled water sample was taken from a pond selected at random from each 1 km survey site where present (a size constraint was used with a pond defined as "a body of standing water 25 m$^2$ to 2 ha in area which usually holds water for at least four months of the year"). The sample was sent to the laboratories at the UK Centre for Ecology and Hydrology, Lancaster. The samples were analysed according to accredited methods as described for headwater streams in Table 4.

To calculate pond biological quality, the method of the Freshwater Habitats Trust (Predictive SYstem for Multimetrics -
PSYM) was used. This is a standardised method (Howard, 2002), summarised as follows:

PSYM was developed to provide a method for assessing the biological quality of still waters in England and Wales. The method uses a number of aquatic plant and invertebrate measures (known as metrics), which are combined together to give a single value which represents the waterbody's overall quality status (Williams et al., 1996). Using the method involves the following steps:

1. Simple environmental data are gathered for each waterbody from map or field evidence (area, grid reference, geology etc.).

2. Biological surveys of the plant and animal communities are undertaken and net samples are processed.

3. The biological and environmental data are entered into the PSYM computer programme which:

(i) uses the environmental data to predict which plants and animals should be present in the waterbody if it is un-degraded

(ii) takes the real plant and animal lists and calculates a number of metrics

Finally the programme compares the predicted plant and animal metrics with the real survey metrics to see how similar they are (i.e. how near the waterbody currently is to its ideal/un-degraded state). The metric scores are then combined to provide a single value which summarises the overall ecological quality of the waterbody. Where appropriate, individual metric scores can also be examined to help diagnose the causes of any observed degradation (e.g. eutrophication, metal contamination) (Williams et al., 1996).


### 2.5 Birds

Bird surveys were coordinated by the British Trust for Ornithology (BTO). The survey protocol (Siriwardena and Taylor, 2014) was designed to provide a robust estimate of the total numbers of breeding pairs of birds of each species found in each
1 km survey square and thus of change over time in future surveys, as well as information on the habitat patches in which individuals were recorded. Thus, the results provide information on local abundance and the selection of habitat types, such as areas under Glastir habitat management (Emmett and GMEP team, 2014). The protocol operates at the same spatial scale as

the national BTO/JNCC/RSPB Breeding Bird Survey (BBS) (Harris et al., 2019), but involves more intensive fieldwork, so it provides more accurate measures of local abundance and is more appropriate for surveying smaller samples of squares each

year (60-90, versus thousands), with lower rates of repetition (Emmett and GMEP team, 2014). Measurement of habitat selection at the patch level also represents a finer scale of inference than is available from the BBS, which aggregates birds and habitats at the scale of the 200 m transect section. The field methods used thus incorporated elements of the BTO's previous national bird monitoring scheme, the Common Birds Census (O'Connor, 1990).

The surveys consisted of four (reduced to three from 2015-16 onwards) visits to each square by trained, professional BTO

surveyors (Siriwardena et al., 2020). Surveys were equally spaced through mid-March to mid-July. On each visit, the surveyor walked a route that passed within 50 m of all parts of the survey square to which access had been secured, beginning at around 06:00 and taking up to five hours. Surveys were not conducted in conditions known to affect the detection of birds, i.e. strong winds and more than light rain. The survey route was started in different places on each visit, so that all areas were visited at least once before 08:00. All birds seen or heard were recorded on high-resolution field maps using standard BTO activity

codes. Recording and standardizing route coverage (where surveyors actually walked) was important both between visits and to ensure comparable repeat coverage when squares are revisited (Siriwardena et al., 2020). The method is a distillation of the approach used for the BTO's Common Birds Census between 1962 and 2000 (O'Connor, 1990).

### 2.6. Pollinators


Each of the 300 1 km squares was visited twice (once each in July and August) in one year between 2013 and 2016. Butterfly Conservation (BC) subcontracted nine experienced ecologists to survey 1 km squares across six regions of Wales (Emmett and GMEP team, 2014). A further region was covered by a BC employee. Pollinator surveys focused on three main pollinator groups: butterflies (Lepidoptera: *Rhopalocera*), bees (Hymenoptera: *Apoidea*) and hoverflies (Diptera: *Syrphidae*). Butterflies

were recorded to species level, whilst bees and hoverflies were recorded as groups based on broad differences in morphological features associated with ecological differences. Note that training was critical for identification of these groups, particularly hoverflies. In addition, the abundance of common flowering plant groups (identified at the time of survey) was recorded using the DAFOR-X scale (**D** (Dominant): >30%, **A** (Abundant): 11-30% , **F** (Frequent): 6-10%, **O** (Occasional): 2-5% , **R** (Rare): 0 - 1%, **X** (not seen on route)) (Emmett and GMEP team, 2014).

Survey visits were split into two independent parts: (1) a standardised 2 km transect route through each 1km square, established following the Wider Countryside Butterfly Survey (WCBS) method (Brereton et al., 2011; UKBMS, 2020) which uses Pollard walks (Pollard, 1977), as used in the UK Butterfly Monitoring Scheme (Brereton et al., 2019) and (2) a timed search in a 150 m$^2$ flower-rich area within the square.

The transect route was split into two approximately parallel 1 km routes separated by at least 500 m, and where possible at

least 250 m in from the edge of the square. These routes were subdivided into ten 200 m sections. In each section the number of each butterfly species and bee and hoverfly group within a 5 m$^2$ recording box were recorded while walking the transect



route at a steady pace. The DAFOR-X abundance of key flowering plant groups (selected on the basis of being known to be important plant groups for pollinating insects) was also recorded within the 5 m$^2$ recording box. At the end of the transect walk the weather conditions were recorded: temperature (°C), sunshine (%) and wind speed (Beaufort scale) (Emmett and GMEP team, 2014).

For the timed searches, surveyors identified a 150 m$^2$ flower rich area within the 1 km square. In this area numbers of butterfly species and bee and hoverfly groups (the same groups as for the transect recording) seen within a 20 minute period were counted. Surveyors also recorded which flowering plant group, if any, these pollinators were visiting.

Surveys were only conducted between 10:00 and 16:00, or between 09:30 and 16:30 if > 75 % of the survey area was un-shaded and weather conditions were suitable for insect activity. The criteria for suitable weather were: temperature between 11 and 17 °C with at least 60 % sunshine or above 17 °C regardless of sunshine, and with a wind speed below 5 on the Beaufort scale ("small trees in leaf sway") (Emmett and GMEP team, 2014).

**2.7 Quality Assurance**

In addition to specific measures already described for each element, Department for Environment, Food and Rural Affairs (DEFRA) Joint Codes of Practice (JCoPR) were followed throughout (DEFRA, 2015). The JCoPR sets out standards for the quality of science and the quality of research processes. This helps ensure the aims and approaches of research are robust. It also gives confidence that processes and procedures used to gather and interpret the results of research are appropriate, rigorous, repeatable and auditable.

The laboratories at the UK Centre for Ecology & Hydrology (UKCEH) Lancaster are UKAS (United Kingdom Accreditation Service, https://www.ukas.com/) accredited. UKCEH maintains a Quality Management System across its four sites which is ISO 9001:2015 certified.

**3. Results to date**

The data collected within the field survey have been analysed extensively. One benefit of the structured sampling approach is that the 'Wider Wales' control sample provides an unbiased national assessment of stock and condition of common habitats including woodland, soils, small streams and ponds. The same approach has been used for reporting on stock and condition of British ecosystems since 1978 by the UK Centre for Ecology and Hydrology through the UKCEH Countryside Survey programme (http://www.countrysidesurvey.org.uk/) (Emmett and GMEP team, 2017). By following the same approach for selecting sites and capturing data in the field, GMEP results can be linked to past trends to put the current observations into context. This has many benefits for interpretation of results. For example, a result of 'no change' (based on a comparison between GMEP and UKCEH Countryside Survey data) could be positive if it indicates a long-term decline has now been halted, but could be negative if a previously reported improvement was now stalled (Emmett and GMEP team, 2017). The key



results were reported to the Welsh Government as outlined below. Beyond this, additional work has also been carried out in several areas.

## 3.1 Key findings as reported to Welsh Government


Key findings were reported to the Welsh Government, alongside modelling and other outputs, in a 2017 report (Emmett and GMEP team, 2017) and online (www.gmep.wales). A brief summary of some of these findings, as reported in 2017 is presented as follows. In terms of biodiversity and habitat condition of land in Wales, plant species indicative of good condition were found to be either stable or improving for arable, improved land, broadleaved woodland and 'habitat land' (land not in the

former three categories; mostly neutral grassland and upland habitat types). The condition of blanket bogs is improving, as is the condition of purple moor grass and rush pasture, two priority habitats (Maddock, 2008). These habitats have been targeted for improvement for many years and many actions have been undertaken to support their recovery. The relative importance of restoration practices, pollution reduction, climate and/or rainfall changes still need to be explored. Initial analysis also suggests a recent increase in the area of blanket bog and montane habitats (Emmett and GMEP team, 2017).

GMEP also identified a set of concerns in some national trends. One such concern is the lack of woodland creation, contrary to the ambitious targets of the Welsh Government (Emmett and GMEP team, 2017). While the mean patch size of habitat, including woodland, was found to have increased over the last 30 years, no change was detected in the area of small woodlands (< 0.5 ha). The small amount of area planted within the Glastir scheme by 2017 (3,923 ha) is within the variability of the GMEP sample. Such small woodlands are not currently captured by the National Forest Inventory (Forest Research, 2020) and

are the woodlands most likely to be affected by Glastir (Emmett and GMEP team, 2017). This does not appear to reflect the targets for expansion of woodlands set by the Welsh Government nor exploit the multiple benefits woodlands can bring for biodiversity, carbon sequestration and water regulation. In fact, this lack of progress, combined with increased agricultural activity, has led to an increase in greenhouse gas emissions in Wales (Committee on Climate Change, 2018). However, there has been an increase in plant species indicative of good condition in large broadleaved woodlands over the last 10 years,

suggesting improved management of existing sites (Emmett and GMEP team, 2017).

Regarding soils, topsoil carbon has been stable or has increased in woodland and improved land soils over the last 30 years. Across all land cover types, overall topsoil has become less acidic over the last three decades, with the most likely reason being the large reductions of acidifying pollutants; emission and deposition of acidifying pollutants across the UK peaked in the 1970s. Recently, a small increase in the acidity of topsoil in improved land has been observed. This may be due to the long-

standing decline in lime use combined with continued fertiliser use. A recent loss of topsoil carbon in 'habitat land' has also been observed, driven primarily by a reduction in carbon concentration in acid grassland and heathland. This trend is currently being investigated further by UKCEH through targeted resampling of soils on acid grassland and heathland sites (Emmett and GMEP team, 2017).



Concerning freshwaters, over the last 20 years, new analyses of small stream data from Natural Resources Wales show an
ongoing improvement in invertebrate diversity and nutrient status (Natural Resources Wales, 2016). GMEP sampling of
headwater streams indicates more than 80% have high diversity according to invertebrate indicators. There are an estimated
9.5 to 16 thousand kilometres of headwater streams in Wales and they are a priority conservation habitat for a range of
characteristic plant and animal species. In terms of livestock, 55% of small streams were found to be freely accessible. This
increases the risk of damage to banks and associated raised sediment levels and increases the risk of phosphorus and pathogen
levels. The latter has implications for contamination of shellfish beds, human health and recreation. It should be noted that
some access to stock is essential for exposed river sediment specialist invertebrates (Emmett and GMEP team, 2017).

Only 13% of ponds sampled in GMEP were judged to be in good ecological condition. Ponds are important to the Welsh
landscape because they provide characteristic habitat and biota and support two thirds of all freshwater species (Freshwater
Habitat Trust, 2021). They act as stepping stones for biota to disperse over wide distances while also providing refuges for
wildlife. They are also priority habitats under the EU habitats directive (Maddock, 2008). There is a substantial amount of
pond habitat in Wales, around 57,800 ponds in total. Further analysis is needed to identify the cause of this poor condition
which could include poor creation practice, lag time after pond creation, runoff from adjacent fields etc. Whilst pond numbers
are high, their ecological value seems in question considering the low number in good condition. Better advice concerning
their creation and management appears to be needed (Emmett and GMEP team, 2017).


## 3.2 Additional published work to date

The breadth and quantity of data available from the field survey offers many opportunities for potential analyses, as evidenced
by work undertaken since the end of the field survey in 2016. Data have been used to investigate vegetation species trends
along linear features, incorporating GMEP data with those from the UKCEH Countryside Survey 1990-2007 (Smart et al.,
2017). Results indicated a continuation of a trend towards increased shading and woody cover.  Furthermore, data from GMEP
vegetation quadrats have been combined with plant trait databases and satellite imagery to map net primary productivity across
Wales (Tebbs et al., 2017). GMEP vegetation data have also provided a national benchmark against which to assess bias in the
contemporary National Plant Monitoring Scheme (Pescott et al., 2019).  Similarly, GMEP pollinator surveys have provided a
national benchmark by which to assess the value of Wales' salt marshes for bees (Davidson et al., 2020). More recently, GMEP
data permitted the most comprehensive assessment of pollinator abundance across Wales' habitats to date, revealing key roles
for woodlands, woody linear features and croplands (Alison et al., 2021). Maskell et al. (2019) combined multiple strands of
GMEP data to understand how species richness is distributed across landscapes, exploring relationships between land-use
intensity, habitat heterogeneity and species richness of multiple taxa in order to map and monitor High Nature Value (HNV)
farmland.

GMEP data have been used to investigate the quality and value of landscapes in Wales as a whole, focusing on how different
landscapes are valued by the public, and transferring the methods to landscapes in Iceland (Swetnam et al., 2017; Swetnam





and Korenko, 2019; Swetnam and Tweed, 2018). GMEP mapping data have also been used for accuracy assessment of land cover maps, produced using satellite imagery (Carrasco et al., 2019).

A range of different work concerning soils has been undertaken since 2016. This includes a consideration of differences in soil physicochemical properties across habitats and relative to known thresholds for supporting habitat function (Seaton et al., 2020a). Several key soil properties, such as carbon, nitrogen and pH, were found to be strongly correlated across soils and can be used to create a soils classification. Soil analyses were complemented by microbiological measurements from DNA metabarcoding of specific target genes, deposited with the European Nucleotide Archive (ENA) at EMBL-EBI (Environment

Centre Wales (Bangor University), 2016a, b, c). These results, and the methods used to determine them are presented in  George et al. (2019a), where it was demonstrated that soil microbial and soil animal taxa respond differently to changes in land use and soil type. Animal richness was governed by intensive land use and unaffected by soil properties, while microbial richness was driven by environmental properties across land uses. The efficacy of 18S and ITS1 barcodes in capturing fungal biological and functional diversity has been compared, revealing barcode biases that influenced metrics of functional but not biological

diversity (George et al., 2019b). Investigations of bacterial functional groups in the 16S marker gene dataset showed changes in sulphate-reducing bacterial community across land uses, with highest richness in grasslands (George et al., 2020).

Soil meso-fauna have been described by George et al. (2017), explaining how broad soil meso-fauna groups differed among disparate habitats, with abundances being lowest in arable sites overall, and Collembola and predatory mites being lower in uplands.

In terms of soil particle size, soil textural heterogeneity was found to be positively linked to bacterial richness for the first time (Seaton et al., 2020b) but fungal richness was not directly impacted by soil texture. Both bacterial and fungal community composition were impacted by the textural composition of the soil. Data from GMEP have even contributed to international studies of the macro ecology of soil bacterial communities (Ramirez et al., 2018).

Soil water repellency results are presented in Seaton et al. (2019). They found that soil water repellency affected 92 % of soils

at a national scale across Wales, and that plant and soil microbial community composition strongly influenced repellency. Repellency is associated with bypass flow in soils which can transmit pollutants faster to groundwater. However, Seaton et al. (2019) proposed a mechanism whereby soil biota mediated the association between repellency and many physicochemical stresses.

Work to date focusing on the freshwater data has concentrated on diatoms (Jones et al., 2017). Various data analysis techniques

were used to explore how indices based on diatom assemblages (related to eutrophication and siltation), diatom species, the traits motility, and nutrient affinity responded to a gradient of percentage cover of fine sediment.

**4. Data Availability**

The datasets have been assigned digital object identifiers and users of the data must reference the data as follows:





- Botham, M. et al. (2020). **Insect pollinator and flower data from the Glastir Monitoring and Evaluation Programme, Wales, 2013-2016.** NERC Environmental Information Data Centre. https://doi.org/10.5285/3c8f4e46-bf6c-4ea1-9340-571fede26ee8
- Keith, A.M. et al. (2019). **Topsoil meso-fauna data from the Glastir Monitoring and Evaluation Programme, Wales 2013-2014.** https://doi.org/10.5285/1c5cf317-2f03-4fef-b060-9eccbb4d9c21
- Lebron, I. et al (2020). **Topsoil particle size distribution from the Glastir Monitoring and Evaluation Programme, Wales 2013-2016.** https://doi.org/10.5285/d6c3cc3c-a7b7-48b2-9e61-d07454639656
- Maskell, L.C. et al. (2020**). Landscape and habitat area data from the Glastir Monitoring and Evaluation Programme, Wales 2013-2016.** https://doi.org/10.5285/82c63533-529e-47b9-8e78-51b27028cc7f
- Maskell, L.C. et al. (2020). **Landscape linear feature data from the Glastir Monitoring and Evaluation Programme, Wales 2013-2016.** https://doi.org/10.5285/f481c6bf-5774-4df8-8776-c4d7bf059d40
- Maskell, L.C. et al. (2020). **Landscape point feature data from the Glastir Monitoring and Evaluation Programme, Wales 2013-2016.** https://doi.org/10.5285/9f8d9cc6-b552-4c8b-af09-e92743cdd3de
- Robinson, D.A et al. (2019). **Topsoil physico-chemical properties from the Glastir Monitoring and Evaluation Programme, Wales 2013-2016.** https://doi.org/10.5285/0fa51dc6-1537-4ad6-9d06-e476c137ed09
- Scarlett, P. et al. (2020). **Pond quality metrics from the Glastir Monitoring and Evaluation Programme, Wales 2013-2016**. https://doi.org/10.5285/687b38d3-2278-41a0-9317-2c7595d6b882
- Scarlett, P. et al. (2020). **Headwater stream quality metrics from the Glastir Monitoring and Evaluation Programme, Wales 2013-2016.** NERC Environmental Information Data Centre. https://doi.org/10.5285/e305fa80-3d38-4576-beef-f6546fad5d45
- Siriwardena, G.M. et al. (2020**). Bird counts from the Glastir Monitoring and Evaluation Programme, Wales 2013-2016.** NERC Environmental Information Data Centre. https://doi.org/10.5285/31da0a94-62be-47b3-b76e-4bdef3037360
- Smart, S.M. et al. (2020). **Vegetation plot data from the Glastir Monitoring and Evaluation Programme, Wales 2013-2016.** https://doi.org/10.5285/71d3619c-4439-4c9e-84dc-3ca873d7f5cc

The datasets are available from the UKCEH Environmental Information Data Centre Catalogue (https://eip.ceh.ac.uk/data). Datasets are provided under the terms of the Open Government Licence (http://www.nationalarchives.gov.uk/doc/open-government-licence/version/3/). The metadata are stored in the ISO 19115 (2003) schema (International Organization for Standardization, 2015) in the UK Gemini 2.3 profile (UK GEMINI, https://www.agi.org.uk/agi-groups/standards-committee/uk-gemini/40-gemini/1037-uk-gemini-standard-and-inspire-implementing-rules).

Users of the datasets will find the following field handbooks useful when re-using data (supplied as supporting documentation with the datasets): Landscape mapping (Maskell et al., 2016a; Maskell et al., 2016b), Vegetation and Soils (Smart et al., 2016), Freshwaters (O'Hare et al., 2013), Birds (Siriwardena and Taylor, 2014) and Pollinators (Botham et al., 2014).

**5. Conclusions**

The data recorded during the GMEP field survey provide an invaluable resource for studying the environment in Wales. The data were collected in a statistically robust and quality controlled manner, follow standard, repeatable methods and cover wide spatial scales. Complemented by data from the UKCEH Countryside Survey of Great Britain, trends across Wales can be

assessed, dating back to 1978. As consequence of this, the data present a unique opportunity for inclusion in a wide range of analyses and models. Data gathered within the field survey are complemented by the additional information arising from the programme of GMEP as a whole, which includes qualitative data such as photographs, social science farmer practice survey
data and assessments of historic features.

The intention is that a repeat survey will be undertaken in the near future in order to provide the opportunity to analyse changes in the countryside (see https://erammp.wales/en for the latest updates).

High level questions and tasks deserving further analysis include investigations of drivers of change, such as looking at evidence for change in the stock and condition of individual broad habitats, exploring the reasons for the finding of decreased
topsoil carbon in 'habitat land' and the increased acidity in improved land, and investigating how the spatial and temporal trends observed in soil, vegetation, pollinators, birds and water are linked. Also of interest would be how climate change and air pollution signals might be distinguished from changes in land management, linked to economic drivers (Emmett and GMEP team, 2017).

Of wider interest to the public might be work to identify the relationship between the area and condition of our natural resources
as indicated by the GMEP survey and the health and well-being of the wider population.

The GMEP data could be exploited to provide an assessment of the general condition of designated lands benchmarked against average national trends (for example, to determine whether soil condition above or below that of the national average). This could assist towards more integrated working for new regulatory frameworks and incentive schemes.

It is expected that the GMEP data, and also modelling and knowledge gained during the programme, will be invaluable when
it comes to tackling new issues such as the United Kingdom's withdrawal from the European Union, particularly in assisting the Welsh Government in developing of new regulatory frameworks and incentive schemes.

### *Author Contribution*

CMW and JA drafted the manuscript and created the figures, with contributions from all co-authors. Authors were each involved in the following areas of work: data management, CMW and JA; vegetation and habitats, SMS, LCM and LRN;
butterflies, RH; birds, GMS; pollinators, MSB; chemical analysis of samples, PK; soils, DAR, PBLG, FMS, AMK and IL; freshwaters, FE and PS; integrated data analyses, SJ. PAH was responsible for the statistical design of the survey and the project informatics strategy. RAG and AB managed the field surveys. BW managed the GMEP project and BAE was the principal investigator. JS oversaw the project.

### *Competing interests*
The authors declare that they have no conflict of interest.



*Acknowledgements*

We thank the approximately 60 members of the GMEP field survey teams who contributed to collecting all the data. We thank the soil processing teams, Gaynor Barrett, Heather Carter, Simon Creer, Beverley Dodd, Andrew Fitton, Claudia Giampieri,
Rob Griffiths, Steve Hughes, Alex Hunt, Davey Jones, Delphine Lallias, Rachel Marshall, David Nuñez, Manisha Patel, Gloria Pereira, Binoti Tanna, and Nicola Thompson. For diatom identification, we thank Bowburn Consultancy, Durham (http://www.bowburn-consultancy.co.uk/) and for freshwater invertebrate identification, we thank APEM Ltd (https://www.apemltd.co.uk/).

We thank the bird survey team, George Tordoff, Rachel Taylor and Mike Edwards. For data management, we thank Shaun
Astbury, Mike Brown, Jane Hall, Katrina Sharps, John Watkins and Simon Wright. For additional survey elements regarding landscapes and historic features, we thank Ruth Swetnam, and Ian Halfpenney and team. For botanical quality assurance, we thank Hilary Wallace of Ecological Surveys (Bangor) Ltd.

We thank Anthea Owen for obtaining permissions to survey from approximately 1500 landowners across Wales. We thank all of the landowners who granted permission to survey on their land, without which the survey could not have taken place.

We also thank the GMEP Stakeholder Group for valuable advice and support.

The research was funded by the Welsh Government through the Glastir Monitoring and Evaluation Programme (GMEP). Contract reference: C147/2010/11. NERC/UK Centre for Ecology & Hydrology (UKCEH Projects: NEC04780/NEC05371/NEC05782).

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
