# Peer review of "Integrated ecological monitoring in Wales: the Glastir Monitoring and Evaluation Programme field survey"

_Earth System Science Data, 2021_

## Referee Comment (RC1)

Thanks to CEH for sharing data from a massive systematic ecological monitoring effort. Agree about magnitude: easily the most comprehensive survey in Wales. As usual for CEH products, well-organised and easily accessible.

I have concerns having to do with impact and uncertainties. About impact, the authors list monitoring the impact of Glastir activities as a primary motivation. But in fact they present no evidence that they monitor or could monitor any outcome of any local individual activity; not their purpose and in fact precluded by their sample design? The survey data have inherent value; why list them as relevant to Glastir-funded activities if readers and data providers both know the irrelevance of these data to that effort? If the proposal that funded this work justified its efforts based on ability to detect impact of specific albeit as-yet unspecified remediation activities, the careful necessary systematic sample design precludes such detection?

Authors offer a second (worthy) motivation: to quantify status and trends. Good, agreed. But particularly trends require uncertainties; value of such-and-such parameter at such-and-such date differs significantly (or not) from same value measured at a different date. Even water chemistry data, some of which (PO4) fall below LOD (limits of detection), have no associated uncertainties. One understands why uncertainties for binary presence-absence data or species lists might prove challenging (but others have addressed these issues?), but - in my extensive but not exhaustive perusal - no file had a +/- uncertainty column. One understands, given this wide range of parameters, that a single encompassing uncertainty (e.g. $\pm$ 95% CI) will not suffice, but reader never finds any uncertainty estimates while authors apparently avoid the topic? ESSD readers expect and require better detail?

Emphasize - data have enormous independent value, not least because of consistency in sample design, parameter choice, quality control, and skill of execution with previous and on-going English, Scottish, etc. surveys (also by CEH). But as a monitoring tool for Glastir impacts? Not likely.

I suspect I understand their caution, but how can one read any description of monitoring ecosystems of Wales without encountering the word 'sheep'. In my direct experience, including time in Snowden, the country was and remains extensively and thoroughly 'sheep-burnt'. Perhaps mention of "livestock" (line 393) or "gazing animals" (line 127) allows authors to allude to sheep without actually mentioning them as the dominant land surface modifier? Any Glastir-funded monitoring effort must carefully follow Glastir expectations, language and protocols or (more cynically) measure only ecosystem features not impacted by sheep? From other reports we learn that Wales houses "10 million" sheep, that 75% of Welsh land is "devoted" to livestock, about negative impacts on vegetation, soil compaction, water quality, etc. From their avian-focussed viewpoint, UK RSPB's State of Nature report found that "60% of animal and plant species in Wales have declined over the last 50 years and 31% have declined strongly with farming practices being blamed for loss of habitats." Does that report and countless others overstate or miss key factors? If authors intend these data to provide "an unbiased national assessment of stock and condition of common habitats" (line 347), and understanding that careful description of data gathering must precede subsequent analysis, results reported so far seem to fit the general characterization (Section 3.1) of 'no change, 'not much deterioration', or not 'as much progress as hoped'. I recognize confusing difficult-to-navigate lines between Glastir funding for monitoring and rigorous national ecological monitoring, but the present project as defined here, wanting to have both, actually satisfies neither?

One final comment related to question of uncertainties: authors rely almost exclusively on internal technical reports not available to most potential data users. Most references refer to technical reports, of UK govt or especially Glastir or CEH. Very few references listed here come from science journals, even fewer from open science journals? Rare good examples George et al., Wood et al. (Note that authors have ESSD abbreviated differently among several Wood et

al. references.) For documents not easily available (see below), we need them included as part of metadata for this activity.

Repeat: excellent data easily accessible and skillfully managed. Questions or concerns from this reviewer have mostly to do with overstatement or mis-statement of intent and impact!

Specific comments:

In Table1, X plots, 200 m2 subsequently reduced to 4 m2? Funding or personnel limitation, but no discussion of impact on data?

Cores thaw during posting? (e.g. line 181)

Peat only mentioned once (in Table 2 methods for SOC (ii)). (Peat related to blanket bogs?) Peat mining represents a substantial ecosystem disturbance?

Emmett and GMEP team 2014, 2017 cited several times, evidently key documents in terms of information, approach, organization, but simply not available? Certainly not available to this reader. Make those full documents available as part of GMEP metadata, on specific CEH GMEP landing page?

Concern, which CEH must have addressed in prior ESSD publications, about reliance on ESRI and ArcGIS, a proprietary software not available to many ESSD readers. These authors to assure that full-function open access alternatives (e.g QGIS) exist in all cases?

---

## Referee Comment (RC2)

**General comments:**

The manuscript '*Integrated ecological monitoring in Wales: the Glastir Monitoring and Evaluation Programme field survey*' describes the set-up and survey protocols of the Glastir Monitoring and Evaluation Programme (GMEP). The surveys are carried out at 300 stratified-random sampled 1 km square test sites across Wales. For assessing the impact of agri-environmental interventions, representative indicators are surveyed which are vegetation, land cover and land use, soil parameters, freshwater, birds and insect pollinators.

This paper mainly focuses on the **survey instructions and the key raw data sets** gained during the performed field surveys. It only provides a rough overview on the already performed data analyses and the results within that programme.

Comprehensive monitoring programmes such as GMEP are extremely valuable and welcome in order to *identify as well as quantify changes in biodiversity of farmland*. In general, for interested external researchers, it is difficult or even impossible to get access to the data sets obtained from nationally conducted research programmes for further data analyses. In this context, I very much appreciate providing open access data sets such as the GMEP data. The GMEP data are presented online in a user-friendly and well-organised way including all key information - field manuals included - required by the user. For a suggestion on data presentation see technical comments below.

Valid modelling of data can only be carried out, if data sets in high quality are available. In this context, comprehensive field data collection is of main priority in biodiversity research although in most cases, there are only tight budgets provided for this issue. I consider the obligation for data users to register and to cite the original data source a valuable step for traceability of data use. I appreciate the idea of a rolling monitoring in the programme which enables studying a maximum number of test sites, while keeping the costs for the monitoring scheme low. Moreover, I consider the splitting into 'Wider Wales component' (baseline estimation) and 'Target Component' (priority areas and aims) to be a useful approach.

Unfortunately, I am not familiar with the specific challenges of monitoring in the UK. In general, monitoring programmes are challenging to be set up, as only limited components/parameters can be investigated and therefore, monitoring programmes can never be complete. In my opinion, the selection of indicators in GMEP is well targeted to reflect biodiversity at different levels. If possible, also grasshoppers could be included in the survey in future as they have already proven to be a useful indicator for farmland connectivity and quality. Dealing with monitoring issues we unfortunately have to accept incompleteness and limitations in the surveys procedure. Maybe the application of classical monitoring tools and approaches in combination with new supporting techniques could help to provide a broader data spectrum in future.

Trend analyses are in general difficult to interpret, as the data of the annual survey rounds are also influenced by non-standardised parameters such as the prevailing weather conditions, natural population fluctuations, etc. In any case, utmost caution is required when interpreting ecological/species trends. Consequently, we always face uncertainties when dealing with the issue. However, it is important to include and consider all parameters that could have an impact on biodiversity (including sheep grazing and other farmland practices) in the analyses. To improve the present state of biodiversity, the main drivers in the monitored region need to be identified.

In several passages, the manuscript reads like *a detailed survey protocol*. I would suggest shortening the detailed descriptions of the fieldwork, and better refer to already existing published protocols/manuals – which are already made available online with GMEP data sets.

The GMEP programme is very comprehensive and as I am not familiar with it, it is not possible for me to comment on and evaluate the significance/suitability/value of the data presented here throughout the programme. Therefore, I have focused my review mainly on the data collection protocols and procedures presented in the manuscript.

**Specific comments:**

The survey procedures on the whole are well conceptualized. In my opinion, the following aspects should be specified and clarified.

1) *Figure 1*: There are no test sites in some areas of the land classes (e.g. light green). What is the reason for this? Is that mentioned area not used as a farmland?

2) *Figure 2*: Were the data collection plots e.g. hedges selected randomly?

3) *Page 6*: It is stated that linear features may contain gaps of up to 20 m (page 6). In this case, should they not better be considered as two linear elements?

4) I think that a size of point features of 20x20 m is too large. Most of single trees will not be considered in that case.

5) I wonder about the rough categorisation of some landscape elements, e.g. urban. In this category also streets are included. Does this category only include sealed roads or also gravelled roads, or do the latter fall within the category 'boundaries', since they have to be evaluated differently from an ecological point of view?

6) How was plant cover estimated? According to an estimation scale, e.g. Braun-Blanquet (1964)?

7) I also think that eDNA samples for soil analyses may serve as a supporting tool in biodiversity monitoring. In which way were the DNA metabarcoding analyses carried out in detail?

8) *Table 2*: Text passages could also be presented in a separate methods chapter or *via* reference to applied soil analyses protocols. Maybe only variations/adaptations should be addressed in the manuscript in detail.

9) Concerning the transect route of butterflies: Does a standardised transect route through each 1 km square mean that the location and direction of the transect were the same in all surveyed test squares?

10) Weather conditions were recorded at the end of the transect walk. Are there any predefined conditions that must be followed when starting the surveys, as in general, it is the case with butterfly surveys?

11) Pollinators, page 18: I would start the last text passage of this chapter with 'For the timed searches, surveyors identified…' In my opinion, this is more logical, because it is not clear to the reader whether there are no preconditions in the butterfly survey that should be considered before starting the survey (see 10).

12) page 19/363: According to which aspects did the plant species improve or be stable? Please specify the statement.

The manuscript mainly focuses on the data collection procedure and the broadness of available data sets in GMEP. An overview of the monitoring results is given, but it does not go into much detail regarding the analyses that could be/were carried out using these data sets. **Yes, there is major**

**potential for further analyses.** I for example, miss analyses calculating biodiversity indices, patch size, landscape complexity, connectivity, corridor effects, etc., which are all essential for the assessment of biodiversity status and change in farmland. Also aspects such as land management (e.g. farmland practice) including conventional and organics farming also considering soil components are not addressed in the manuscript.

**Technical errors:**

Data presentation: I have checked some of the online provided species lists. I think the lists would be more user-friendly, if all the information was not summarized in one column, but in separate ones with own column headings.

---

## Author Comment (AC1)

*Response: Many thanks for all the constructive comments regarding our manuscript which we have endeavoured to address as follows:*

*1) comments from Referees, (2) author's response, (3) author's changes in manuscript.*

*Response to Anonymous Referee #1*

1) Thanks to CEH for sharing data from a massive systematic ecological monitoring effort. Agree about magnitude: easily the most comprehensive survey in Wales. As usual for CEH products, well-organised and easily accessible.
**2) Many thanks for the positive comments.**

**1)** I have concerns having to do with impact and uncertainties. About impact, the authors list monitoring the impact of Glastir activities as a primary motivation. But in fact they present no evidence that they monitor or could monitor any outcome of any local individual activity; not their purpose and in fact precluded by their sample design? The survey data have inherent value; why list them as relevant to Glastir-funded activities if readers and data providers both know the irrelevance of these data to that effort? If the proposal that funded this work justified its efforts based on ability to detect impact of specific albeit as-yet unspecified remediation activities, the careful necessary systematic sample design precludes such detection?
**2) The design of the GMEP survey is aimed at both monitoring the effects of the Glastir scheme and also trends in the wider countryside, with sites 'Targeted' at Glastir priority areas, and also sites located in 'Wider Wales'. It is acknowledged the design is not aimed at monitoring local individual activities or interventions, for example on a farm-scale basis. Rather, the aim of the design is to detect broad trends at a large-scale national level. Additionally, by repeating the surveys, it is intended that long-term trends may be identified. It is recognised that combining the data with additional third party data could be an advantage to potential users depending on their intended use. An example of the integration with third party data can already be seen in Maskell et al., 2019.**
**3) Line 51, insert 'at a national scale' to clarify the scope of the Glastir evaluation, and lines 52, insert 'in the longer term' to emphasise the potential of the long term monitoring aspect of the survey.**

*Maskell, L.C., Botham, M., Henrys, P., Jarvis, S., Maxwell, D., Robinson, D.A., Rowland, C.S., Siriwardena, G., Smart, S., Skates, J. and Tebbs, E.J., 2019. Exploring relationships between land use intensity, habitat heterogeneity and biodiversity to identify and monitor areas of High Nature Value farming. Biological Conservation, 231, pp.30-38.*

**1)** Authors offer a second (worthy) motivation: to quantify status and trends. Good, agreed. But particularly trends require uncertainties; value of such-and-such parameter at such-and-such date differs significantly (or not) from same value measured at a different date. Even water chemistry data, some of which ($PO_4$) fall below LOD (limits of detection), have no associated uncertainties. One understands why uncertainties for binary presence-absence data or species lists might prove challenging (but others have addressed these issues?), but - in my extensive but not exhaustive perusal - no file had a +/- uncertainty column. One understands, given this wide range of parameters, that a single encompassing uncertainty (e.g. + 95% CI) will not suffice, but reader never finds any uncertainty estimates while authors apparently avoid the topic? ESSD readers expect and require better detail?
**2) Of course analysed trends and summaries require some indication of uncertainties, for example as shown in published results (e.g. https://gmep.wales/biodiversity). However, the data sets in question here are the raw data, as collected in the field (or laboratory) and so do not have any**

calculated uncertainty values.  We acknowledge there is variability in terms of surveyor recording and effort, and also in laboratory measurements.  In order to counter this, stringent quality control procedures have been in place throughout the survey as described in the text relating to each survey element, in terms of surveyor training and repeated samplings.

**3) Line 346, insert 'and the results and associated uncertainties are publicly available via https://gmep.wales/data-findings.' to emphasise availability of results and findings, including associated uncertainties.**

**1)** Emphasize - data have enormous independent value, not least because of consistency in sample design, parameter choice, quality control, and skill of execution with previous and ongoing English, Scottish, etc. surveys (also by CEH). But as a monitoring tool for Glastir impacts? Not likely.

**2) See comment above.  Again, we emphasize the broad scale, potentially long-term nature of the surveys. It is possible to use the data to undertake analyses such as 'where there is more uptake of intervention X, do we see an improvement in water quality, soil health, etc.?' Again, it is acknowledged that combining the data with additional third party data could be an advantage to potential users, depending on their individual interest.**

**3) See above.**

**1)** I suspect I understand their caution, but how can one read any description of monitoring ecosystems of Wales without encountering the word 'sheep'. In my direct experience, including time in Snowden, the country was and remains extensively and thoroughly 'sheep-burnt'. Perhaps mention of "livestock" (line 393) or "gazing animals" (line 127) allows authors to allude to sheep without actually mentioning them as the dominant land surface modifier? Any Glastir funded monitoring effort must carefully follow Glastir expectations, language and protocols or (more cynically) measure only ecosystem features not impacted by sheep? From other reports we learn that Wales houses "10 million" sheep, that 75% of Welsh land is "devoted" to livestock, about negative impacts on vegetation, soil compaction, water quality, etc. From their avian-focussed viewpoint, UK RSPB's State of Nature report found that "60% of animal and plant species in Wales have declined over the last 50 years and 31% have declined strongly with farming practices being blamed for loss of habitats." Does that report and countless others overstate or miss key factors? If authors intend these data to provide "an unbiased national assessment of stock and condition of common habitats" (line 347), and understanding that careful description of data gathering must precede subsequent analysis, results reported so far seem to fit the general characterization (Section 3.1) of 'no change, 'not much deterioration', or not 'as much progress as hoped'. I recognize confusing difficult-to-navigate lines between Glastir funding for monitoring and rigorous national ecological monitoring, but the present project as defined here, wanting to have both, actually satisfies neither?

**2) Yes, sheep are indeed an important factor when considering land use in Wales.  However, this paper is primarily focused on the data and outcomes specifically gathered from the GMEP field survey.  Whilst limited data on sheep were collected during the survey, the GMEP/ERAMMP teams have undertaken comprehensive modelling activities, which very much consider the role of sheep (for example https://erammp.wales/sites/default/files/ERAMMP_Rpt-26_QuickStart-2_Small_Sectors_v1.0_en.pdf ).  This is beyond the scope of this paper, however.  If data users were interested in sheep, the field data do provide opportunities for additional analyses if desired.**

**The RSPB report, published in 2013 before the GMEP survey does state 'Due to a lack of suitable data, we were only able to present quantitative trends for about 5% of the UK's species, and when we look at a smaller scale, the problem becomes even greater. As a result, although we report the best available data here for Wales, the picture is far from complete – we simply don't have sufficient**

knowledge to make a robust, quantitative assessment of the state of nature in Wales.' The GMEP data aims to fill this gap.

Again, the emphasis is on broad, national trends, and establishing a comprehensive baseline on which to base future trend analyses.

3) Insert line 358, 'Results from the field survey are also complemented by outputs from other parts of the GMEP Programme, such as modelling (for example Emmet et al., 2017)'. Also see above.

1) One final comment related to question of uncertainties: authors rely almost exclusively on internal technical reports not available to most potential data users. Most references refer to technical reports, of UK govt or especially Glastir or CEH. Very few references listed here come from science journals, even fewer from open science journals? Rare good examples George et al., Wood et al. (Note that authors have ESSD abbreviated differently among several Wood et al. references.) For documents not easily available (see below), we need them included as part of metadata for this activity.

2) Yes, we agree this is an issue, however, one of the primary aims and motivations of this paper is exactly to address this by publishing this key information in a peer-reviewed journal rather than the many available contract reports. The contract reports referred to are openly available on the NERC Open Research Archive (NORA), a long-term repository for NERC and NERC related centre (i.e. UKCEH) outputs.

3) We will ensure that all reports are clearly signposted to the openly available copies in the manuscript. We will ensure the abbreviations of ESSD are consistent.

1) Repeat: excellent data easily accessible and skillfully managed. Questions or concerns from this reviewer have mostly to do with overstatement or mis-statement of intent and impact!

2) Many thanks. We will attempt to address the issues of overstatement or mis-statement of intent and impact as summarised under the different comment sections.

*1) Specific comments:*

1) In Table1, X plots, 200 m2 subsequently reduced to 4 m2? Funding or personnel limitation, but no discussion of impact on data?

2) Unfortunately the change was enforced due to time constraints. There is no major impact on the data as there is nothing to suggest that large plots are better than small ones for the intended analyses – the plots still provide consistent data at that particular level of survey. Larger plots have been found to be optimal within woodlands (Wood et al., 2015), and the large plots were maintained in woodland habitats throughout GMEP. The larger, nested plots do give additional scope to analyse data at different levels, and potentially compare across different third party datasets. However, this is an additional benefit, and does not mean the smaller plots are not fit for purpose in terms of the analyses undertaken to date.

3) Clarify that large plots were maintained in woodlands in Table 1.

1) Cores thaw during posting? (e.g. line 181)

2) Cores are not frozen until arrival at the laboratory.

3) We will ensure this is clear in the text.

1) Peat only mentioned once (in Table 2 methods for SOC (ii)). (Peat related to blanket bogs?) Peat mining represents a substantial ecosystem disturbance?

2) Yes, peat is an important consideration. As part of the GMEP programme, a new peat map for Wales was produced (https://doi.org/10.5285/58139ce6-63f9-4444-9f77-fc7b5dcc00d8 ).

**However, again this was not specifically part of the field survey, and so again is out of the scope of this paper.**

**1)** Emmett and GMEP team 2014, 2017 cited several times, evidently key documents in terms of information, approach, organization, but simply not available? Certainly not available to this reader. Make those full documents available as part of GMEP metadata, on specific CEH GMEP landing page?

**2) The contract reports referred to are openly available on the NERC Open Research Archive (NORA), a long-term repository for NERC and NERC related centre (i.e. UKCEH) outputs.**

**3) We will ensure that all reports are clearly signposted to the openly available copies in the manuscript, and will consider adding to the metadata of specific data sets if appropriate.**

**1)** Concern, which CEH must have addressed in prior ESSD publications, about reliance on ESRI and ArcGIS, a proprietary software not available to many ESSD readers. These authors to assure that full-function open access alternatives (e.g QGIS) exist in all cases?

**2) It is important to emphasize the distinction with data collection and data re-use. UKCEH made the decision to use ESRI products due to the requirements of the complex, bespoke and large-scale nature of the data collection, and the support required from ESRI.  However, this does not mean that users of the data require access to ESRI software.  EIDC endeavour to ensure that data are available in non-proprietary formats, hence all of the GMEP datasets with a spatial nature are available in .csv formats, or shapefiles which may be opened and analysed in a range of different software, include open source options.**

**3) We will ensure this is clear in the text. Line 461: 'The datasets are available in non-proprietary formats..'**

***Response to Anonymous Referee #2***

**1) General comments:**

The manuscript '*Integrated ecological monitoring in Wales: the Glastir Monitoring and Evaluation Programme field survey*' describes the set-up and survey protocols of the Glastir Monitoring and Evaluation Programme (GMEP). The surveys are carried out at 300 stratified-random sampled 1 km square test sites across Wales. For assessing the impact of agri-environmental interventions, representative indicators are surveyed which are vegetation, land cover and land use, soil parameters, freshwater, birds and insect pollinators.

This paper mainly focuses on the **survey instructions and the key raw data sets** gained during the performed field surveys. It only provides a rough overview on the already performed data analyses and the results within that programme.

Comprehensive monitoring programmes such as GMEP are extremely valuable and welcome in order to *identify as well as quantify changes in biodiversity of farmland*. In general, for interested external researchers, it is difficult or even impossible to get access to the data sets obtained from nationally conducted research programmes for further data analyses. In this context, I very much appreciate providing open access data sets such as the GMEP data. The GMEP data are presented online in a user-friendly and well-organised way including all key information - field manuals included - required by the user. For a suggestion on data presentation see technical comments below.

**2) Many thanks for the positive comments.**

**1)** Valid modelling of data can only be carried out, if data sets in high quality are available. In this context, comprehensive field data collection is of main priority in biodiversity research although in most cases, there are only tight budgets provided for this issue. I consider the obligation for data users to register and to cite the original data source a valuable step for traceability of data use. I appreciate the idea of a rolling monitoring in the programme which enables studying a maximum number of test sites, while keeping the costs for the monitoring scheme low. Moreover, I consider the splitting into 'Wider Wales component' (baseline estimation) and 'Target Component' (priority areas and aims) to be a useful approach.

**2) Many thanks for the positive comments.**

**1)** Unfortunately, I am not familiar with the specific challenges of monitoring in the UK. In general, monitoring programmes are challenging to be set up, as only limited components/parameters can be investigated and therefore, monitoring programmes can never be complete. In my opinion, the selection of indicators in GMEP is well targeted to reflect biodiversity at different levels. If possible, also grasshoppers could be included in the survey in future as they have already proven to be a useful indicator for farmland connectivity and quality. Dealing with monitoring issues we unfortunately have to accept incompleteness and limitations in the surveys procedure. Maybe the application of classical monitoring tools and approaches in combination with new supporting techniques could help to provide a broader data spectrum in future.

**2) The idea of including grasshoppers is interesting, thank you for the suggestion. We acknowledge that new supporting techniques will be a useful addition in the future, in the particular the increased use of earth observation for habitat identification which is under ongoing consideration, for example Henrys, PA, Jarvis, SG. Integration of ground survey and remote sensing derived data: Producing robust indicators of habitat extent and condition. Ecol Evol. 2019; 9: 8104– 8112. https://doi.org/10.1002/ece3.5376**

**1)** Trend analyses are in general difficult to interpret, as the data of the annual survey rounds are also influenced by non-standardised parameters such as the prevailing weather conditions, natural population fluctuations, etc. In any case, utmost caution is required when interpreting ecological/species trends. Consequently, we always face uncertainties when dealing with the issue. However, it is important to include and consider all parameters that could have an impact on biodiversity (including sheep grazing and other farmland practices) in the analyses. To improve the present state of biodiversity, the main drivers in the monitored region need to be identified.

**2) Yes, indeed. To date, much of the analysis undertaken has focused on Glastir outcomes, the results of which are shown here: https://gmep.wales/data-findings The data may be analysed in relation to many drivers and, for example, contribute to assessments of national resources such as the State of Natural Resources reports, 2016 and 2020.**

**https://naturalresources.wales/evidence-and-data/research-and-reports/the-state-of-natural-resources-report-assessment-of-the-sustainable-management-of-natural-resources/?lang=en**

**https://naturalresources.wales/evidence-and-data/research-and-reports/state-of-natural-resources-report-sonarr-for-wales-2020/?lang=en**

**1)** In several passages, the manuscript reads like *a detailed survey protocol*. I would suggest shortening the detailed descriptions of the fieldwork, and better refer to already existing published protocols/manuals – which are already made available online with GMEP data sets.

The GMEP programme is very comprehensive and as I am not familiar with it, it is not possible for me to comment on and evaluate the significance/suitability/value of the data presented here throughout the programme. Therefore, I have focused my review mainly on the data collection protocols and procedures presented in the manuscript.

**2) Thank you, this is a good suggestion. We were trying to be as comprehensive as possible but agree that some of the protocol sections could be shortened.**

**3) Cuts to text: Lines 155+, Table 2, Lines 199+, Lines 228+, Table 4.**

**Specific comments:**

**1)** The survey procedures on the whole are well conceptualized. In my opinion, the following aspects should be specified and clarified.

**1)** 1) *Figure 1*: There are no test sites in some areas of the land classes (e.g. light green). What is the reason for this? Is that mentioned area not used as a farmland?

**2) As the sites were selected randomly within each land class, this has led to the appearance that some areas contain no test sites. In the case of Targeted sites, it is the case that no targeted areas were selected in those areas; in the case of the Wider Wales sites, it is entirely due to chance. The key requisite is that the land class is represented by an adequate number of sites in order to be statistically valid when scaling the data up. This is described in lines 84-91, but is perhaps not clear.**

**3) We will clarify 94+ 'The number of 1 km squares randomly sampled and sited from each land class was proportional to the area of that land class in Wales.'**

**1)** 2) *Figure 2*: Were the data collection plots e.g. hedges selected randomly?

**2) Yes, the hedges are selected randomly, in relation to the X plot. Whilst we allude to the positioning by referring to the Countryside Survey protocols, we agree this could be clarified in the text.**

**3) We will amend table 1 to clarify this.**

**1)** 3) *Page 6*: It is stated that linear features may contain gaps of up to 20 m (page 6). In this case, should they not better be considered as two linear elements?

**2) The rules for recording hedges were agreed in consultation with hedgerow experts (now Hedgelink https://hedgelink.org.uk/) when standardising the UKCEH methodology (for example as part of the UK Countryside Survey). Long woody linear features may have gaps up to 20m, but a gap bigger than 20m would result in the remaining two sections being recorded separately. 20m was arrived at as the cut-off point for pragmatic reasons and knowledge/experience of the lengths of gaps commonly present in the countryside. If gaps were, for example 10m, then this would create a lot more work for surveyors in terms of creating many more separate linear features with common attributes.**

**3) We will add in this justification into lines 135. '(a rule agreed with hedgerow experts when compiling methods for UKCEH Countryside Survey (Maskell, 2008)).'**

**1)** 4) I think that a size of point features of 20x20 m is too large. Most of single trees will not be considered in that case.

**2) There seems to be some confusion here – the text states 'Point features are individual landscape elements that occupy less than an area of 20 m × 20 m'. Therefore, anything larger than 20x20m is mapped as an area, otherwise it is represented as a point. This ensures that single trees are mapped as points.**

**3) We will amend the text to ensure this is clear. Line 139: 'landscape elements that occupy an area of less than 20 m × 20 m'**

**1)** 5) I wonder about the rough categorisation of some landscape elements, e.g. urban. In this category also streets are included. Does this category only include sealed roads or also gravelled roads, or do the latter fall within the category 'boundaries', since they have to be evaluated differently from an ecological point of view?
**2) The urban category does include all types of streets, roads and tracks (if wider than 5m). However, this landscape survey is primarily a survey of the countryside, hence the majority of land categorised as urban is excluded from the reported analyses. None of the other survey elements (i.e. soils, plants, freshwater, pollinators, birds) are collected from land categorised as urban. However, it is important to note that in terms of scaling up areal habitat data from a 1km square, for statistical reasons, it is essential to know the proportion of each 1km square assigned to urban categories (in addition to the proportion of sea and refused access land). The estimate of habitats must be based on the proportion of surveyed land, not the proportion of the full 1km squares. Otherwise, the non-urban habitats would be overestimated. This includes a mask of non-surveyed urban land (i.e. large towns and cities), but also surveyed urban land (occurring in surveyed squares with <75% urban land). This is discussed in detail in Barr, C. J.; Bunce, R. G. H.; Clarke, R. T.; Fuller, R. M.; Furse, M. T.; Gillespie, M. K.; Groom, G. B.; Hallam, C. J.; Hornung, M.; Howard, D. C.; Ness, M. J.. 1993 Countryside Survey 1990: main report. (Countryside 1990 vol.2). London, Department of the Environment, 174pp, Appendix 3 and also D.C. Howard, J.W. Watkins, R.T. Clarke, C.L. Barnett, G.J. Stark, Estimating the extent and change in Broad Habitats in Great Britain, Journal of Environmental Management, Volume 67, Issue 3, 2003, Pages 219-227, https://doi.org/10.1016/S0301-4797(02)00175-5 .**

**1)** 6) How was plant cover estimated? According to an estimation scale, e.g. Braun-Blanquet (1964)?
**2) Cover estimates were made to the nearest 5 % for all species reaching at least an estimated 5 % cover. Surveyors make a percentage cover estimation, not using any particular scale. Surveyors calibrate their estimations when being trained. Whilst it is recognised this is not an exact science, the cover estimates give an important record of the relative abundances of species within the plots.**
**3) Line 157 – 'Cover estimates were made to the nearest 5 % for all species reaching at least an estimated 5 % cover.'**

**1)** 7) I also think that eDNA samples for soil analyses may serve as a supporting tool in biodiversity monitoring. In which way were the DNA metabarcoding analyses carried out in detail?
**2) Yes, agreed. The methods used to carry out the DNA metabarcoding are presented in George, P. B., Lallias, D., Creer, S., Seaton, F. M., Kenny, J. G., Eccles, R. M., Griffiths, R. I., Lebron, I., Emmett, B. A., and Robinson, D. A.: Divergent national-scale trends of microbial and animal biodiversity revealed across diverse temperate soil 565 ecosystems, Nature communications, 10, 1-11, doi:https://doi.org/10.1038/s41467-019-09031-1, 2019a. In the interests of brevity, we have not detailed these methods in this text, as they are described in detail in George et al., 2019.**
**3)** Amend text 'The methods for this area of work is outlined in George et al., 2019a'

**1)** 8) *Table 2*: Text passages could also be presented in a separate methods chapter or *via* reference to applied soil analyses protocols. Maybe only variations/adaptations should be addressed in the manuscript in detail.
**2) Thank you for the suggestion.**
**3) As described previously, we will endeavour to reduce and shorten the protocol sections.**

**1)** 9) Concerning the transect route of butterflies: Does a standardised transect route through each 1 km square mean that the location and direction of the transect were the same in all surveyed test squares?

**2) Transect routes are standardised as 2km long, comprising of two parallel 1-km long survey lines across the 1km$^2$. The survey lines should run N-S or E-W or as close as possible, and should be subdivided into ten continuous 200m sections numbered 1-10.  The location and direction are not exactly the same in all test squares due to a number of reasons including: barriers such as fences, railways, roads and rivers; allowable access permission; the avoidance of urban habitats.  This is the standard UK Butterfly Monitoring Scheme method.**

**3) Insert line 321: 'Flexibility in the route was allowed based on the presence of barriers such as roads and railways, urban areas and refused access permission.'**

**1)** 10) Weather conditions were recorded at the end of the transect walk. Are there any predefined conditions that must be followed when starting the surveys, as in general, it is the case with butterfly surveys?

**2) As stated, 'Surveys were only conducted between 10:00 and 16:00, or between 09:30 and 16:30 if > 75 % of the survey area was un-shaded and weather conditions were suitable for insect activity. The criteria for suitable weather were: temperature between 320 11 and 17 °C with at least 60 % sunshine or above 17 °C regardless of sunshine, and with a wind speed below 5 on the Beaufort scale ("small trees in leaf sway")'.  Lines 319-322. There were no other areas of the survey dependant on predefined conditions.**

**3)  The conditions are stated in lines 319-322.**

**1)** 11) Pollinators, page 18: I would start the last text passage of this chapter with 'For the timed searches, surveyors identified…' In my opinion, this is more logical, because it is not clear to the reader whether there are no preconditions in the butterfly survey that should be considered before starting the survey (see 10).

**2) Good suggestion, we will amend the text.**

**3) Update lines ~326+.**

**1)** 12) page 19/363: According to which aspects did the plant species improve or be stable? Please specify the statement.

**2) Good suggestion, thank you.  Text clarified as below.**

**3) 'In terms of biodiversity and habitat condition of land in Wales, high-quality habitat plant indicator species (positive Common Standard Monitoring (CSM) Species, JNCC, 2021) were found to be either stable or increasing for arable, improved land, broadleaved woodland and 'habitat land' (land not in the former three categories; mostly neutral grassland and upland habitat types).**

**(JNCC, 2021, https://jncc.gov.uk/our-work/common-standards-monitoring/ )**

**1)** The manuscript mainly focuses on the data collection procedure and the broadness of available data sets in GMEP. An overview of the monitoring results is given, but it does not go into much detail regarding the analyses that could be/were carried out using these data sets. **Yes, there is major**

**potential for further analyses.** I for example, miss analyses calculating biodiversity indices, patch size, landscape complexity, connectivity, corridor effects, etc., which are all essential for the assessment of biodiversity status and change in farmland. Also aspects such as land management (e.g. farmland practice) including conventional and organics farming also considering soil components are not addressed in the manuscript.

**2) Thank you for the suggestions of ideas for additional analyses. Many analyses have already been undertaken by members of the GMEP team, for example those published at https://gmep.wales/data-findings (including patch size, biodiversity indices and connectivity), as mentioned above, and some of the suggestions are out of the scope of these datasets, such as organic farming.  Due to space limitations, it is not possible to comprehensively describe all of these analyses already undertaken, but yes, it is valuable to point out that these results have been published.**

**There is so much potential for many types of analyses that it is difficult to suggest a select few within the limits of this paper.  There are already some suggestions in the conclusion, for example, investigations of drivers of change, such as looking at evidence for change in the stock and condition of individual broad habitats, exploring the reasons for the finding of decreased topsoil carbon in 'habitat land' and the increased acidity in improved land, and investigating how the spatial and temporal trends observed in soil, vegetation, pollinators, birds and water are linked.  It is a good idea to highlight that data can be combined with third party information for more extensive analyses.**

**3) Amend line 349: 'The data collected within the field survey have been analysed extensively and the results and associated uncertainties are publicly available via https://gmep.wales/data-findings'**

**Amend line 483: 'Combining the field datasets with information from third party sources would provide additional opportunities for more extensive analyses.'**

**1) Technical errors:**

Data presentation: I have checked some of the online provided species lists. I think the lists would be more user-friendly, if all the information was not summarized in one column, but in separate ones with own column headings.

**2)  Thank you for the suggestion.  Different people have different preferences as to how the data are presented, according to their intended use.  We have endeavoured to present the data in a way that is most flexible, allowing the user to restructure in the manner suggested if preferred.**